# SIMPLE INITIALIZATION AND PARAMETRIZATION OF SINUSOIDAL NETWORKS VIA THEIR KERNEL BANDWIDTH

**Filipe de Avila Belbute-Peres**
Carnegie Mellon University
Pittsburgh, PA
`filiped@cs.cmu.edu`

**J. Zico Kolter**
Carnegie Mellon University & Bosch Center for AI
Pittsburgh, PA
`zkolter@cs.cmu.edu`

## ABSTRACT

Neural networks with sinusoidal activations have been proposed as an alternative to networks with traditional activation functions. Despite their promise, particularly for learning implicit models, their training behavior is not yet fully understood, leading to a number of empirical design choices that are not well justified. In this work, we first propose a simplified version of such sinusoidal neural networks, which allows both for easier practical implementation and simpler theoretical analysis. We then analyze the behavior of these networks from the neural tangent kernel perspective and demonstrate that their kernel approximates a low-pass filter with an adjustable bandwidth. Finally, we utilize these insights to inform the sinusoidal network initialization, optimizing their performance for each of a series of tasks, including learning implicit models and solving differential equations.

## 1 INTRODUCTION

Sinusoidal networks are neural networks with sine nonlinearities, instead of the traditional ReLU or hyperbolic tangent. They have been recently popularized, particularly for applications in implicit representation models, in the form of SIRENs (Sitzmann et al., 2020). However, despite their popularity, many aspects of their behavior and comparative advantages are not yet fully understood. Particularly, some initialization and parametrization choices for sinusoidal networks are often defined arbitrarily, without a clear understanding of how to optimize these settings in order to maximize performance.

In this paper, we first propose a simplified version of such sinusoidal networks, that allows for easier implementation and theoretical analysis. We show that these simple sinusoidal networks can match and outperform SIRENs in implicit representation learning tasks, such as fitting videos, images and audio signals. We then analyze sinusoidal networks from a neural tangent kernel (NTK) perspective (Jacot et al., 2018), demonstrating that their NTK approximates a low-pass filter with adjustable bandwidth. We confirm, through an empirical analysis this theoretically predicted behavior also holds approximately in practice. We then use the insights from this analysis to inform the choices of initialization and parameters for sinusoidal networks. We demonstrate we can optimize the performance of a sinusoidal network by tuning the bandwidth of its kernel to the maximum frequency present in the input signal being learned. Finally, we apply these insights in practice, demonstrating that "well tuned" sinusoidal networks outperform other networks in learning implicit representation models with good interpolation outside the training points, and in learning the solution to differential equations.

## 2 BACKGROUND AND RELATED WORK

**Sinusoidal networks.** Sinusoidal networks have been recently popularized for implicit modelling tasks by sinusoidal representation networks (SIRENs) (Sitzmann et al., 2020). They have also been evaluated for physics-informed learning, demonstrating promising results in a series of domains (Raissi et al., 2019b; Song et al., 2021; Huang et al., 2021b;a; Wong et al., 2022). Among the benefits of such networks is the fact that the mapping of inputs through an (initially) random linear layer followed by a sine function is mathematically equivalent to a transformation to a random Fourier basis, rendering them close to networks with Fourier feature transforms (Tancik et al., 2020; Rahimi & Recht,

2007), and possibly able to address spectral bias (Basri et al., 2019; Rahaman et al., 2019; Wang et al., 2021). Sinusoidal networks also have the property that the derivative of their outputs is given simply by another sinusoidal network, due to the fact that the derivative of sine function is a phase-shifted sine.

**Neural tangent kernel.** An important prior result to the neural tangent kernel (NTK) is the neural network Gaussian process (NNGP). At random initialization of its parameters $\theta$, the output function of a neural network of depth $L$ with nonlinearity $\sigma$, converges to a Gaussian process, called the NNGP, as the width of its layers $n_1, \ldots, n_L \to \infty$. (Neal, 1994; Lee et al., 2018). This result, though interesting, does not say much on its own about the behavior of *trained* neural networks. This role is left to the NTK, which is defined as the kernel given by $\Theta(x, \tilde{x}) = \langle \nabla_\theta f_\theta(x), \nabla_\theta f_\theta(\tilde{x}) \rangle$. It can be shown that this kernel can be written out as a recursive expression involving the NNGP. Importantly, Jacot et al. (2018) demonstrated that, again as the network layer widths $n_1, \ldots, n_L \to \infty$, the NTK is (1) deterministic at initialization and (2) constant throughout training. Finally, it has also been demonstrated that under some assumptions on its parametrization, the output function of the trained neural network $f_\theta$ converges to the kernel regression solution using the NTK (Lee et al., 2020; Arora et al., 2019). In other words, under certain assumptions the behavior of a trained deep neural network can be modeled as kernel regression using the NTK.

**Physics-informed neural networks.** Physics-informed neural networks (Raissi et al., 2019a) are a method for approximating the solution to differential equations using neural networks (NNs). In this method, a neural network $\hat{u}(t, x; \theta)$, with learned parameters $\theta$, is trained to approximate the actual solution function $u(t, x)$ to a given partial differential equation (PDE). Importantly, PINNs employ not only a standard "supervised" data loss, but also a physics-informed loss, which consists of the differential equation residual $\mathcal{N}$. Thus, the training loss consists of a linear combination of two loss terms, one directly supervised from data and one informed by the underlying differential equations.

## 3 SIMPLE SINUSOIDAL NETWORKS

There are many details that complicate the practical implementation of current sinusoidal networks. We aim to propose a simplified version of such networks in order to facilitate theoretical analysis and practical implementation, by removing such complications.

As an example we can look at SIRENs, which have their layer activations defined as $f_l(x) = \sin(\omega(W_l x + b_l))$. Then, in order to cancel the $\omega$ factor, layers after the first one have their weight initialization follow a uniform distribution with range $[-\frac{\sqrt{6/n}}{\omega}, \frac{\sqrt{6/n}}{\omega}]$, where $n$ is the size of the layer. Unlike the other layers, the first layer is sampled from a uniform distribution with range $[-1/n, 1/n]$.

We instead propose a simple sinusoidal network, with the goal of formulating an architecture that mainly amounts to substituting its activation functions by the sine function. We will, however, keep the $\omega$ parameter, since (as we will see in future analyses) it is in fact a useful tool for allowing the network to fit inputs of diverse frequencies. The layer activation equations of our simple sinusoidal network, with parameter $\omega$, are defined as

$$\begin{aligned} f_1(x) &= \sin(\omega\,(W_1 x + b_1)), \\ f_l(x) &= \sin(W_l x + b_l), \quad l > 1. \end{aligned} \tag{1}$$

Finally, instead of utilizing a uniform initialization as in SIRENs (with different bounds for the first and subsequent layers), we propose initializing all parameters in our simple sinusoidal network using a default Kaiming (He) normal initialization scheme. This choice not only greatly simplifies the initialization scheme of the network, but it also facilitates theoretical analysis of the behavior of the network under the NTK framework, as we will see in Section 4.

**Analysis of the initialization scheme.** The initialization scheme proposed above differs from the one implemented in SIRENs. We will now show that this particular choice of initialization distribution preserves the variance of the original proposed SIREN initialization distribution. As a consequence, the original theoretical justifications for its initialization scheme still hold under this activation, namely that the distribution of activations across layers are stable, well-behaved and shift-invariant. Due to space constraints, proofs are presented in Appendix A. Moreover, we also demonstrate empirically that these properties are maintained in practice.

**Lemma 1.** *Given any c, for $X \sim \mathcal{N}\left(0, \frac{1}{3}c^2\right)$ and $Y \sim \mathcal{U}(-c, c)$, we have $\mathrm{Var}[X] = \mathrm{Var}[Y] = \frac{1}{3}c^2$.*

This simple Lemma and relates to Lemma 1.7 in Sitzmann et al. (2020), showing that the initialization we propose here has the same variance as the one proposed for SIRENs. Using this result we can translate the result from the main Theorem 1.8 from Sitzmann et al. (2020), which claims that the SIREN initialization indeed has the desired properties, to our proposed initialization:[1]

*For a uniform input in $[-1, 1]$, the activations throughout a sinusoidal network are approximately standard normal distributed before each sine non-linearity and arcsine-distributed after each sine non-linearity, irrespective of the depth of the network, if the weights are distributed normally, with mean $0$ and variance $\frac{2}{n}$, where $n$ is a layer's fan-in.*

**Empirical evaluation of initialization scheme.** To empirically demonstrate the proposed simple initialization scheme preserves the properties from the SIREN initialization scheme, we perform the same analysis performed by Sitzmann et al. (2020). We observe that the distribution of activations matches the predicted normal (before the non-linearity) and arcsine (after the non-linearity) distributions, and that this behavior is stable across many layers. These results are reported in detail in the Appendix B.

## 3.1 COMPARISON TO SIREN

In order to demonstrate our simplified sinusoidal network has comparable performance to a standard SIREN, in this section we reproduce the main results from Sitzmann et al. (2020). Table 1 compiles the results for all experiments. In order to be fair, we compare the simplified sinusoidal network proposed in this chapter with both the results directly reported in Sitzmann et al. (2020), and our own reproduction of the SIREN results (using the same parameters and settings as the original). We can see from the numbers reported in the table that the performance of the simple sinusoidal network proposed in this chapter matches the performance of the SIREN in all cases, in fact surpassing it in most of the experiments. Qualitative results are presented in Appendix C.

It is important to note that this is not a favorable setting for simple sinusoidal networks, given that the training durations were very short. The SIREN favors quickly converging to a solution, though it does not have as strong asymptotic behavior. This effect is likely due to the multiplicative factor applied to later layers described in Section 3. We observe that indeed in almost all cases we can compensate for this effect by simply increasing the learning rate in the Adam optimizer (Kingma & Ba, 2014).

Finally, we observe that besides being able to surpass the performance of SIREN in most cases in a short training regimen, the simple sinusoidal network performs even more strongly with longer training. To demonstrate this, we repeated some experiments from above, but with longer training durations. These results are shown in Table 4 in Appendix C.

## 4 NEURAL TANGENT KERNEL ANALYSIS

In the following we derive the NTK for sinusoidal networks. This analysis will show us that the sinusoidal networks NTK is approximately a low-pass filter, with its bandwidth directly defined by $\omega$. We support these findings with an empirical analysis as well in the following section. Finally, we demonstrate how the insights from the NTK can be leveraged to properly "tune" sinusoidal networks to the spectrum of the desired signal. Full derivations and extensive, detailed analysis are left to Appendix D.

The NTK for a simple sinusoidal network with a single hidden layer is presented in the theorem below. The NTK for siren with 1 and 6 hidden layers are shown in Figure 1.

**Theorem 2.** *Shallow SSN NTK. For a simple sinusoidal network with one hidden layer $f^{(1)} : \mathbb{R}^{n_0} \to \mathbb{R}^{n_2}$ following Definition 1, its neural tangent kernel (NTK), as defined in Theorem 6, is given by*

$$\Theta^{(1)}(x, \tilde{x}) = \frac{1}{2} \left( \omega^2 \left( x^T \tilde{x} + 1 \right) + 1 \right) e^{-\frac{\omega^2}{2} \|x - \tilde{x}\|_2^2}$$

$$- \frac{1}{2} \left( \omega^2 \left( x^T \tilde{x} + 1 \right) - 1 \right) e^{-\frac{\omega^2}{2} \|x + \tilde{x}\|_2^2} e^{-2\omega^2} + \omega^2 \left( x^T \tilde{x} + 1 \right) + 1.$$

---

[1] We note that despite being named Theorem 1.8 in Sitzmann et al. (2020), this result is not fully formal, due to the Gaussian distribution being approximated without a formal analysis of this approximation. Additionally, a CLT result is employed which assumes infinite width, which is not applicable in this context. We thus refrain from calling our equivalent result a theorem. Nevertheless, to the extent that the argument is applicable, it would still hold for our proposed initialization, due to its dependence solely on the variance demonstrated in Lemma 1 above.

Table 1: Comparison of the simple sinusoidal network and SIREN results, both directly from Sitzmann et al. (2020) and from our own reproduced experiments. Values above the horizontal center line are peak signal to noise ratio (PSNR), values below are mean squared error (MSE), except for SDF which uses a composite loss. [†]Audio experiments utilized a separate learning rate for the first layer.

| Experiment | Simple Sinusoidal Network | SIREN [paper] | SIREN [ours] |
|---|---|---|---|
| Image | **50.04** | 49 (approx.) | 49.0 |
| Poisson (Gradient) | **39.66** | 32.91 | 38.92 |
| Poisson (Laplacian) | **20.97** | 14.95 | 20.85 |
| Video (cat) | **34.03** | 29.90 | 32.09 |
| Video (bikes) | **37.4** | 32.88 | 33.75 |
| Audio (Bach)[†] | $1.57 \cdot 10^{-5}$ | $\mathbf{1.10 \cdot 10^{-5}}$ | $3.28 \cdot 10^{-5}$ |
| Audio (counting)[†] | $\mathbf{3.17 \cdot 10^{-4}}$ | $3.82 \cdot 10^{-4}$ | $4.38 \cdot 10^{-4}$ |
| Helmholtz equation | $\mathbf{5.94 \cdot 10^{-2}}$ | – | $5.97 \cdot 10^{-2}$ |
| SDF (room) | **12.99** | – | 14.32 |
| SDF (statue) | 6.22 | – | **5.98** |

We can see that for values of $\omega > 2$, the second term quickly vanishes due to the $e^{-2\omega^2}$ factor. This leaves us with only the first term, which has a Gaussian form. Due to the linear scaling term $x^T \tilde{x}$, this is only approximately Gaussian, but the approximation improves as $\omega$ increases. We can thus observe that this kernel approximates a Gaussian kernel, which is a low-pass filter, with its bandwidth defined by $\omega$. Figure 1 presents visualizations for NTKs for the simple sinusoidal network, compared to a (scaled) pure Gaussian with variance $\omega^{-2}$, showing there is a close match between the two.

If we write out the NTK for networks with more than one hidden layer, it quickly becomes un-interpretable due to the recursive nature of the NTK definition (see Appendix D). However, as shown empirically in Figure 1, these kernels are still approximated by Gaussians with variance $\omega^{-2}$.

We also observe that the NTK for a SIREN with a single hidden layer is analogous, but with a $\mathrm{sinc}$ form, which is also a low-pass filter.

**Theorem 3.** *Shallow SIREN NTK. For a single hidden layer SIREN $f^{(1)} : \mathbb{R}^{n_0} \to \mathbb{R}^{n_2}$ following Definition 1, its neural tangent kernel (NTK), as defined in Theorem 6, is given by*

$$\Theta^{(1)}(x, \tilde{x}) = \frac{c^2}{6} \left( \omega^2 \left( x^T \tilde{x} + 1 \right) + 1 \right) \prod_{j=1}^{n_0} \mathrm{sinc}(c\,\omega\,(x_j - \tilde{x}_j))$$

$$- \frac{c^2}{6} \left( \omega^2 \left( x^T \tilde{x} + 1 \right) - 1 \right) e^{-2\omega^2} \prod_{j=1}^{n_0} \mathrm{sinc}(c\,\omega\,(x_j + \tilde{x}_j)) + \omega^2 \left( x^T \tilde{x} + 1 \right) + 1.$$

For deeper SIREN networks, the kernels defined by the later layers are in fact Gaussian too, as discussed in Appendix D. This leads to an NTK that is approximated by a product of a sinc function and a Gaussian. These SIREN kernels are also presented in Figure 1.

## 5 EMPIRICAL ANALYSIS

As shown above, neural tangent kernel theory suggests that sinusoidal networks work as low-pass filters, with their bandwidth controlled by the parameter $\omega$. In this section, we demonstrate empirically that we can observe this predicted behavior even in real sinusoidal networks. For this experiment, we generate a $512 \times 512$ monochromatic image by super-imposing two orthogonal sinusoidal signals, each consisting of a single frequency, $f(x, y) = \cos(128\pi x) + \cos(32\pi y)$. This function is sampled in the domain $[-1, 1]^2$ to generate the image on the left of Figure 2.

To demonstrate what we can expect from applying low-pass filters of different bandwidths to this signal, we perform a discrete Fourier transform (DFT), cut off frequencies above a certain value, and perform an inverse transform to recover the (filtered) image. The MSE of the reconstruction, as a

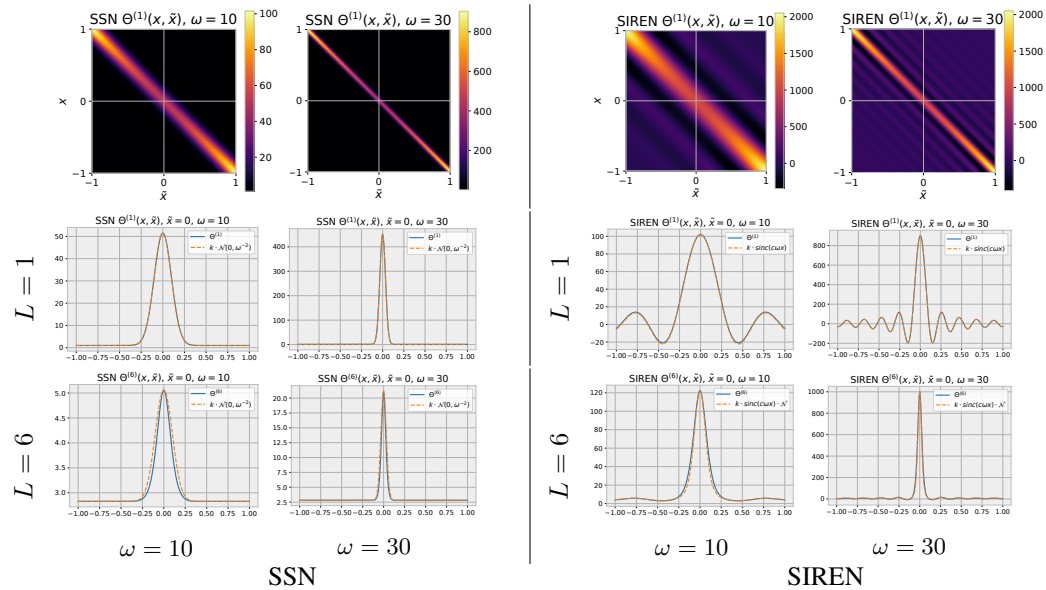

Figure 1: The NTK for SIREN and SSN at different $\omega$. *Top:* Kernel values for pairs $(x, \tilde{x}) \in [-1, 1]^2$. *Bottom:* Slice at fixed $\tilde{x} = 0$. SSN plots show a superimposed Gaussian kernel with variance $\omega^{-2}$ scaled to match the max and min values of the NTK. Similarly, SIREN plots show a sinc function.

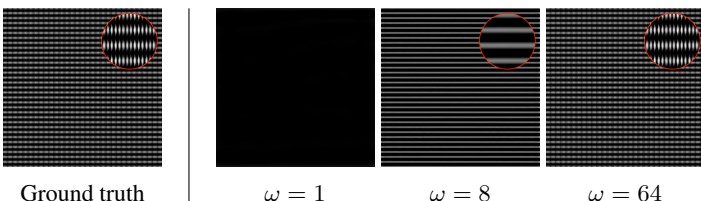

Figure 2: *Left:* The test signal used to analyze the behavior of sinusoidal networks. It is created from two orthogonal single frequencies, $f(x, y) = \cos(128\pi x) + \cos(32\pi y)$. *Right:* Examples of the reconstructed signal from networks with different $\omega$, demonstrating each of the loss levels in Figure 3.

function of the cutoff frequency, is shown in Figure 3. We can see that due to the simple nature of the signal, containing only two frequencies, there are only three loss levels. If indeed the NTK analysis is correct and sinusoidal networks act as low-pass filters, with bandwidth controlled by $\omega$, we should be able to observe similar behavior with sinusoidal networks with different $\omega$ values. We plot the final training loss and training curves for sinusoidal networks with different $\omega$ in Figure 3. We can observe, again, that there are three consistent loss levels following the magnitude of the $\omega$ parameter, in line with the intuition that the sinusoidal network is working as a low-pass filter. This is also observable in Figure 2, where we see example reconstructions for networks of various $\omega$ values after training.

However, unlike with the DFT low-pass filter (which does not involve any learning), we see in Figure 3 that during training some sinusoidal networks shift from one loss level to a lower one. This demonstrates that sinusoidal networks differ from true low-pass filters in that their weights can change, which implies that the bandwidth defined by $\omega$ also changes with learning. We know the weights $W_1$ in the first layer of a sinusoidal network, given by $f_1(x) = \sin(\omega \cdot W_1^T x + b_1)$, will change with training. Empirically, we observed that the spectral norm of $W_1$ increases throughout training for small $\omega$ values. We can interpret that as the overall magnitude of the term $\omega \cdot W_1^T x$ increasing, which is functionally equivalent to an increase in $\omega$ itself. In Figure 3, we observe that sinusoidal networks with smaller values of $\omega$ take a longer time to achieve a lower loss (if at all). Intuitively, this happens because, due to the effect described above, lower $\omega$ values require a larger increase in magnitude by the weights $W_1$. Given that all networks were trained with the same learning rate, the ones with a smaller $\omega$ require their weights to move a longer distance, and thus take more training steps to achieve a lower loss.

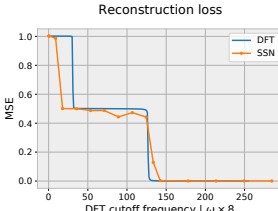 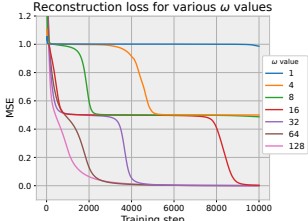

Figure 3: *Left:* Reconstruction loss for different cutoff frequencies for a low-pass filter applied to Figure 2. The three loss levels reflect the 2 frequencies present in the simple signal: 128 and 32. Superimposed is the final loss, after fitting the same input image with sinusoidal networks, for different values of $\omega$ (scaled to align with the frequencies). *Right:* Training curves for the sinusoidal networks. The three loss levels reflect the 2 frequencies present in the simple signal, and demonstrate that the sinusoidal network is indeed acting as a low-pass filter with bandwidth defined by $\omega$.

## 6 TUNING $\omega$

As shown in the previous section, though the bandwidth of a network can change throughout training, the choice of $\omega$ still influences how easily and quickly (if at all) it can learn a given signal. The value of the $\omega$ parameter is thus crucial for the learning of the network. Despite this fact, in SIRENs, for example, this value is not adjusted for each task (except for the audio fitting experiments), and is simply set empirically to an arbitrary value. In this section, we seek to justify a proper initialization for this parameter, such that it can be chosen appropriately for each given task.

Moreover, it is often not the case that we simply want to fit only the exact training samples but instead want to find a good interpolation (*i.e.*, generalize well). Setting $\omega$ too high, and thus allowing the network to model frequencies that are much larger than the ones present in the actual signal is likely to cause overfitting. This is demonstrated empirically in Figure 4.

Consequently, we want instead to tune the network to the highest frequency present in the signal. However, we do not always have the knowledge of what is the value of the highest frequency in the true underlying signal of interest. Moreover, we have also observed that, since the network learns and its weights change in magnitude, that value in fact changes with training. Therefore, the most we can hope for is to have a good heuristic to guide the choice of $\omega$. Nevertheless, having a reasonable guess for $\omega$ is also likely sufficient for good performance, precisely due to the ability of the network to adapt during training and compensate for a possibly slightly suboptimal choice.

**Choosing $\omega$ from the Nyquist frequency.** One source of empirical information on the relationship between $\omega$ and the sinusoidal network's "learnable frequencies" is the previous section's empirical analysis. Taking into account the scaling, we can see from Fig. 3 that around $\omega = 16$ the network starts to be able to learn the full signal (freq. 128). We can similarly note that at about $\omega = 4$ the sinusoidal network starts to be able to efficiently learn a signal with frequency 32, but not the one with frequency 128. This scaling suggests a heuristic of setting $\omega$ to about $1/8$ of the signal's maximum frequency.

For natural signals, such as pictures, it is common for frequencies up to the Nyquist frequency of the discrete sampling to be present. We provide an example for the "camera" image we have utilized so far in Figure 23 in Appendix E, where we can see that the reconstruction loss through a low-pass filter continues to decrease significantly up to the Nyquist frequency for the image resolution.

In light of this information, analyzing the choices of $\omega$ for the experiments in Section 3.1 again suggests that $\omega$ should be set around $1/8$ of the Nyquist frequency of the signal. These values of $\omega$ are summarized in Table 2 in the "Fitting $\omega$" column. For example, the image fitting experiment shows that, for an image of shape $512 \times 512$ (and thus Nyquist frequency of 256 for each dimension), this heuristic suggests an $\omega$ value of $256/8 = 32$, which is the value found to work best empirically through search. We find similar results for the audio fitting experiments. The audio signals used in the audio fitting experiment contained approximately $300,000$ and $500,000$ points, and thus maximum frequencies of approximately $150,00$ and $250,000$. This suggests reasonable values for $\omega$ of $18,750$ and $31,250$, which are close to the ones found empirically to work well. In examples such as the video fitting experiments, in which each dimension has a different frequency, it is not completely

clear how to pick a single $\omega$ to fit all dimensions. This suggests that having independent values of $\omega$ for each dimension might be useful for such cases, as discussed in the next section.

Finally, when performing the generalization experiments in Section 7, we show the best performing $\omega$ ended up being half the value of the best $\omega$ used in the fitting tasks from Section 3.1. This follows intuitively, since for the generalization task we set apart half the points for training and the other half for testing, thus dividing the maximum possible frequency in the training sample in half, providing further evidence of the relationship between $\omega$ and the maximum frequency in the input signal.

**Multi-dimensional $\omega$.** In many problems, such as the video fitting and PDE problems, not only is the input space multi-dimensional, it also contains time and space dimensions (which are additionally possibly of different shape). This suggests that employing a multi-dimensional $\omega$, specifying different frequencies for each dimension might be beneficial. In practice, if we employ a scaling factor $\lambda = \begin{bmatrix} \lambda_1 & \lambda_2 & \dots & \lambda_d \end{bmatrix}^T$, we have the first layer of the sinusoidal network given by

$$f_1(x) = \sin(\omega\,(W_1\,(\lambda \odot x) + b_1)) = \sin(W_1\,(\Omega \odot x) + \omega b_1), \tag{2}$$

where $\Omega = \begin{bmatrix} \lambda_1\omega & \lambda_2\omega & \dots & \lambda_d\omega \end{bmatrix}^T$ works as a multi-dimensional $\omega$. In the following experiments, we employ this approach to three-dimensional problems, in which we have time and differently shaped space domains, namely the video fitting and physics-informed neural network PDE experiments. For these experiments, we report the $\omega$ in the form of the (already scaled) $\Omega$ vector for simplicity.

**Choosing $\omega$ from available information** Finally, in many problems we do have some knowledge of the underlying signal we can leverage, such as in the case of inverse problems. For example, let's say we have velocity fields for a fluid and we are trying to solve for the coupled pressure field and the Reynolds number using a physics-informed neural network (as done in Section 7). In this case, we have access to two components of the solution field. Performing a Fourier transform on the training data we have can reveal the relevant spectrum and inform our choice of $\omega$. If the maximum frequency in the signal is lower than the Nyquist frequency implied by the sampling, this can lead to a more appropriate choice of $\omega$ than suggested purely from the sampling.

## 7    Experiments

In this section, we first perform experiments to demonstrate how the optimal value of $\omega$ influences the generalization error of a sinusoidal network, following the discussion in Section 6. After that, we demonstrate that sinusoidal networks with properly tuned $\omega$ values outperform traditional physics-informed neural networks in classic PDE tasks.

### 7.1    Evaluating generalization

We now evaluate the simple sinusoidal network generalization capabilities. To do this, in all experiments in this section we segment the input signal into training and test sets using a checkerboard pattern – along all axis-aligned directions, points alternate between belonging to train and test set. We perform audio, image and video fitting experiments. When performing these experiments, we search for the best performing $\omega$ value for generalization (defined as performance on the held-out points). We report the best values on Table 2. We observe that, as expected from the discussion in Section 6, the best performing $\omega$ values follow the heuristic discussed above, and are in fact half the best-performing value found in the previous fitting experiments from Section 3.1, confirming our expectation. This is also demonstrated in the plot in Figure 4. Using a higher $\omega$ leads to overfitting and poor generalization outside the training points. This is demonstrated in Figure 4, in which we can see that choosing an appropriate $\omega$ value from the heuristics described previously leads to a good fit and interpolation. Setting $\omega$ too high leads to interpolation artifacts, due to overfitting of spurious high-frequency components.

For the video signals, which have different size along each axis, we employ a multi-dimensional $\omega$. We scale each dimension of $\omega$ proportional to the size of the input signal along the corresponding axis.

Table 2: Generalization results and the respective tuned $\omega$ value. Generalization values are mean squared error (MSE). We can observe the best performing $\omega$ for generalization is half the $\omega$ used previously for fitting the full signal due to the fact that this task used half the sample points from previously.

| Experiment | SIREN | SSN | $\omega$ | Fitting $\omega$ |
|---|---|---|---|---|
| Image | $2.76 \cdot 10^{-4}$ | $\mathbf{1.25 \cdot 10^{-4}}$ | 16 | 32 |
| Audio (Bach) | $4.55 \cdot 10^{-6}$ | $\mathbf{3.87 \cdot 10^{-6}}$ | 8,000 | 15,000 |
| Audio (counting) | $1.37 \cdot 10^{-4}$ | $\mathbf{5.97 \cdot 10^{-5}}$ | 16,000 | 32,000 |
| Video (cat) | $3.40 \cdot 10^{-3}$ | $\mathbf{1.76 \cdot 10^{-3}}$ | $\begin{bmatrix} 4 & 8 & 8 \end{bmatrix}$ | 8 |
| Video (bikes) | $2.74 \cdot 10^{-3}$ | $\mathbf{8.79 \cdot 10^{-4}}$ | $\begin{bmatrix} 4 & 4 & 8 \end{bmatrix}$ | 8 |

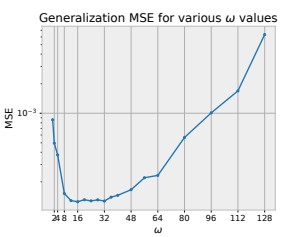

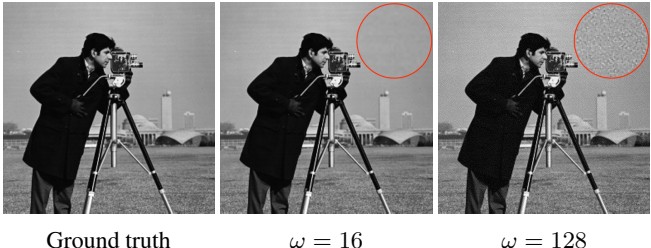

Ground truth      $\omega = 16$      $\omega = 128$

Figure 4: *Left:* Final training loss for different values of $\omega$ in the image fitting generalization experiment. *Right:* Examples of generalization from half the points using sinusoidal networks with different values of $\omega$. Even though both networks achieve equivalent training loss, the rightmost one, with $\omega$ higher than what would be suggested from the Nyquist frequency of the input signal, overfits the data, causing high-frequency noise artifacts in the reconstruction (*e.g.*, notice the sky).

## 7.2 SOLVING DIFFERENTIAL EQUATIONS

Finally, we apply our analysis to physics-informed learning. We compare the performance of simple sinusoidal networks to the $\tanh$ networks that are commonly used for these tasks. Results are summarized in Table 3. Details for the Schrödinger and Helmholtz experiments are presented in Appendix E.

### 7.2.1 BURGERS EQUATION (IDENTIFICATION)

This experiment reproduces the Burgers equation identification experiment from Raissi et al. (2019a). Here we are identifying the parameters $\lambda_1$ and $\lambda_2$ of a 1D Burgers equation, $u_t + \lambda_1 u u_x - \lambda_2 u_{xx} = 0$, given a known solution field. The ground truth value of the parameters are $\lambda_1 = 1.0$ and $\lambda_2 = 0.01/\pi$.

In order to find a good value for $\omega$, we perform a low-pass reconstruction of the solution as before. We can observe in Figure 5 that the solution does not have high bandwidth, with most of the loss being minimized with only the lower half of the spectrum. Note that the sampling performed for the training data ($N = 2,000$) is sufficient to support such frequencies. This suggests an $\omega$ value in the range $8 - 10$. Indeed, we observe that $\omega = 10$ gives the best identification of the desired parameters, with errors of $0.0071\%$ and $0.0507\%$ for $\lambda_1$ and $\lambda_2$ respectively, against errors of $0.0521\%$ and $0.4522\%$ of the baseline. This value of $\omega$ also achieves the lowest reconstruction loss against the known solution, with an MSE of $8.034 \cdot 10^{-4}$. Figure 5 shows the reconstructed solution using the identified parameters.

### 7.2.2 NAVIER-STOKES (IDENTIFICATION)

This experiment reproduces the Navier-Stokes identification experiment from Raissi et al. (2019a). In this experiment, we are trying to identify, the parameters $\lambda_1, \lambda_2$ and the pressure field $p$ of the 2D Navier-Stokes equations given by $\frac{\partial u}{\partial t} + \lambda_1 u \cdot \nabla u = -\nabla p + \lambda_2 \nabla^2 u$, given known velocity fields $u$ and $v$. The ground truth value of the parameters are $\lambda_1 = 1.0$ and $\lambda_2 = 0.01$.

Unlike the 1D Burgers case, in this case the amount of points sampled for the training set ($N = 5,000$) is not high, compared to the size of the full solution volume, and is thus the limiting factor for the bandwidth of the input signal. Given the random sampling of points from the full solution, the

Table 3: Comparison of the sinusoidal network and MLP with tanh non-linearity on PINN experiments from (Raissi et al., 2019a; Sitzmann et al., 2020). Values are percent error relative to ground truth value for each parameter for identification problems and mean squared error (MSE) for inference problems. The Helmholtz experiment is the same from Section 3.1.

| Experiment | Baseline | SSN | $\omega$ | | |
|---|---|---|---|---|---|
| Burgers (Identification) | $[0.0521\%, 0.4522\%]$ | $[\mathbf{0.0071\%}, \mathbf{0.0507\%}]$ | 10 | | |
| Navier-Stokes (Identification) | $[0.0046\%, 2.093\%]$ | $[\mathbf{0.0038\%}, \mathbf{1.782\%}]$ | $[0.6$ | $0.3$ | $1.2]$ |
| Schrödinger (Inference) | $1.04 \cdot 10^{-3}$ | $\mathbf{4.30 \cdot 10^{-4}}$ | 4 | | |
| Helmholtz (Inference) | $5.97 \cdot 10^{-2}$ | $\mathbf{5.94 \cdot 10^{-2}}$ | 16 | | |

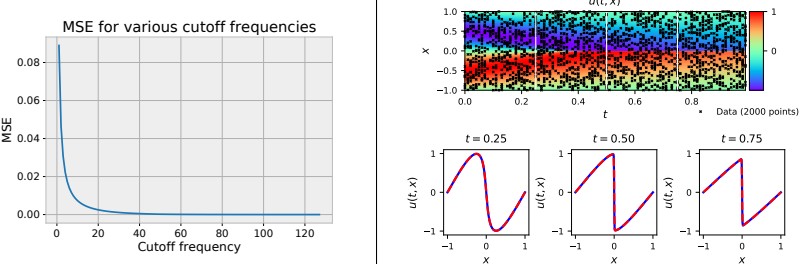

Figure 5: *Left:* Reconstruction loss for different cutoff frequencies for a low-pass filter applied to the solution of the Burgers equation. *Right:* Reconstructed solution of the Burgers equation using the identified parameters with the sinusoidal network, together with the position of the sampled training points.

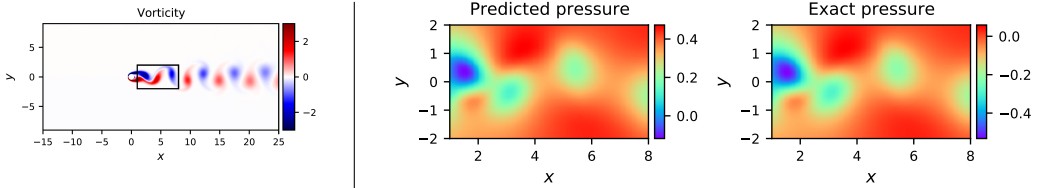

Figure 6: *Left:* One timestep of the ground truth Navier-Stokes solution. The black rectangle indicates the domain region used for the task. *Right:* Identified pressure field for the Navier-Stokes equations using the sinusoidal network. Notice that the identification is only possible up to a constant.

generalized sampling theorem applies. The original solution has dimensions of $100 \times 50 \times 200$. With the $5,000$ randomly sampled points, the average sampling rate per dimension is approximately $17$, on average, corresponding to a Nyquist frequency of approximately $8.5$. Furthermore, given the multi-dimensional nature of this problem, with both spatial and temporal axes, we employ an independent scaling to $\omega$ for each dimension. The analysis above suggests an average $\omega \approx 1$, with the dimensions of the problem suggesting scaling factors of $[0.5 \quad 1 \quad 2]^T$. Indeed, we observe that $\Omega = [0.3 \quad 0.6 \quad 1.2]^T$ gives the best results. With with errors of $0.0038\%$ and $1.782\%$ for $\lambda_1$ and $\lambda_2$ respectively, against errors of $0.0046\%$ and $2.093\%$ of the baseline. Figure 6 shows the identified pressure field. Note that given the nature of the problem, this field can only be identified up to a constant.

## 8 CONCLUSION

In this work, we have present a simplified formulation for sinusoidal networks. Analysis of this architecture from the neural tangent kernel perspective, combined with empirical results, reveals that the kernel for sinusoidal networks corresponds to a low-pass filter with adjustable bandwidth. We leverage this information in order to initialize these networks appropriately, choosing their bandwidth such that it is tuned to the signal being learned. Employing this strategy, we demonstrated improved results in both implicit modelling and physics-informed learning tasks.

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

## A SIMPLE SINUSOIDAL NETWORK INITIALIZATION

We present here the proofs for the initialization scheme of the simple sinusoidal network from Section 3.

**Lemma 4.** *Given any c, for $X \sim \mathcal{N}\left(0, \frac{1}{3}c^2\right)$ and $Y \sim \mathcal{U}\left(-c, c\right)$, we have $\mathrm{Var}[X] = \mathrm{Var}[Y] = \frac{1}{3}c^2$.*

*Proof.* By definition, $\mathrm{Var}[X] = \sigma^2 = \frac{1}{3}c^2$. For $Y$, we know that the variance of a uniformly distributed random variable with bound $[a, b]$ is given by $\frac{1}{12}(b - a)^2$. Thus, $\mathrm{Var}[Y] = \frac{1}{12}(2c)^2 = \frac{1}{3}c^2$. □

**Theorem 5.** *For a uniform input in $[-1, 1]$, the activations throughout a sinusoidal networks are approximately standard normal distributed before each sine non-linearity and arcsine-distributed after each sine non-linearity, irrespective of the depth of the network, if the weights are distributed normally, with mean $0$ and variance $\frac{2}{n}$ with $n$ is the layer's fan-in.*

*Proof.* The proof follows exactly the proof for Theorem 1.8 in Sitzmann et al. (2020), only using Lemma 4 when necessary to show that the initialization proposed here has the same variance necessary for the proof to follow. □

## B EMPIRICAL EVALUATION OF SSN INITIALIZATION

Here we report an empirical analysis the initialization scheme of simple sinusoidal networks, referenced in Section 3. For this analysis we use a sinusoidal MLP with 6 hidden layers of 2048 units, and single-dimensional input and output. This MLP is initialized using the simplified scheme described above. For testing, $2^8$ equally spaced inputs from the range $[-1, 1]$ are passed through the network. We then plot the histogram of activations after each linear operation (before the sine non-linearity) and after each sine non-linearity. To match the original plot, we also plot the 1D Fast Fourier Transform of all activations in a layer, and the gradient of this output with respect to each activation. These results are presented in Figure 8. The main conclusion from this figure is that the distribution of activations matches the predicted normal (before the non-linearity) and arcsine (after the non-linearity) distributions, and that this behavior is stable across many layers. We also reproduced the same result up to 50 layers.

We then perform an additional experiment in which the exact same setup as above is employed, yet the 1D inputs are shifted by a large value (*i.e.*, $x \rightarrow x + 1000$). We the show the same plot as before in Figure 9. We can see that there is essentially no change from the previous plot, which demonstrates the sinusoidal networks shift-invariance in the input space, one of its important desirable properties, as discussed previously.

## C EXPERIMENTAL DETAILS FOR COMPARISON TO SIREN

Below, we present qualitative results and describe experimental details for each experiment. As these are a reproduction of the experiments in Sitzmann et al. (2020), we refer to their details as well for further information.

### C.1 IMAGE

In the image fitting experiment, we treat an image as a function from the spatial domain to color values $(x, y) \rightarrow (r, g, b)$. In the case of a monochromatic image, used here, this function maps instead to one-dimensional intensity values. We try to learn a function $f : \mathbb{R}^2 \rightarrow \mathbb{R}$, parametrized as a sinusoidal network, in order to fit such an image.

Figure 7 shows the image used in this experiment, and the reconstruction from the fitted sinusoidal network. The gradient and Laplacian for the learned function are also presented, demonstrating that higher order derivatives are also learned appropriately.

Table 4: Comparison of the simple sinusoidal network and SIREN on some experiments, with a longer training duration. The specific durations are described below in the details for each experiment. We can see that the simple sinusoidal network has stronger asymptotic performance. Values above the horizontal center line are peak signal to noise ratio (PSNR), values below are mean squared error (MSE). $^{\dagger}$Audio experiments utilized a different learning rate for the first layer, see the full description below for details.

| Experiment | Simple Sinusoidal Network | SIREN [ours] |
|---|---|---|
| Image | **54.70** | 52.43 |
| Poisson (Gradient) | **39.51** | 38.70 |
| Poisson (Laplacian) | **22.09** | 20.82 |
| Video (cat) | **34.64** | 32.26 |
| Video (bikes) | **37.71** | 34.07 |
| Audio (Bach)$^{\dagger}$ | $\mathbf{5.66 \cdot 10^{-7}}$ | $3.02 \cdot 10^{-6}$ |
| Audio (counting)$^{\dagger}$ | $\mathbf{4.02 \cdot 10^{-5}}$ | $6.33 \cdot 10^{-5}$ |

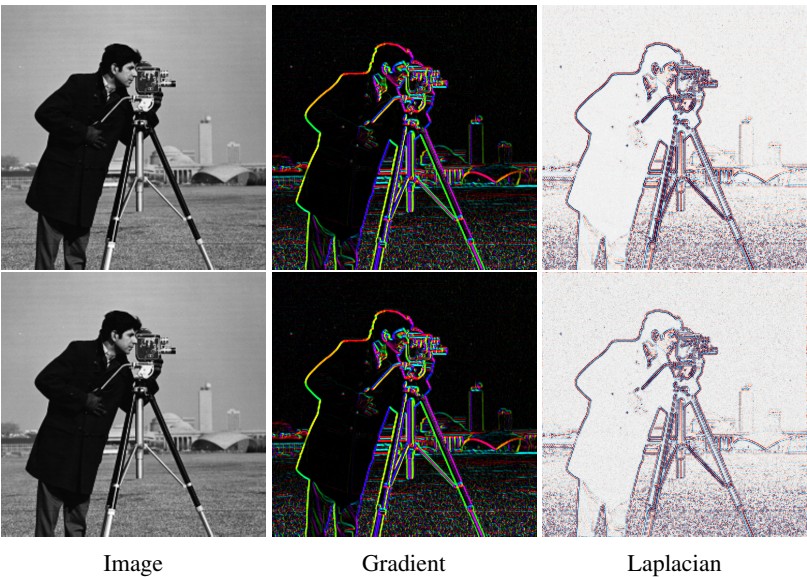

| Image | Gradient | Laplacian |

Figure 7: Top row: Ground truth image. Bottom: Reconstructed with sinusoidal network.

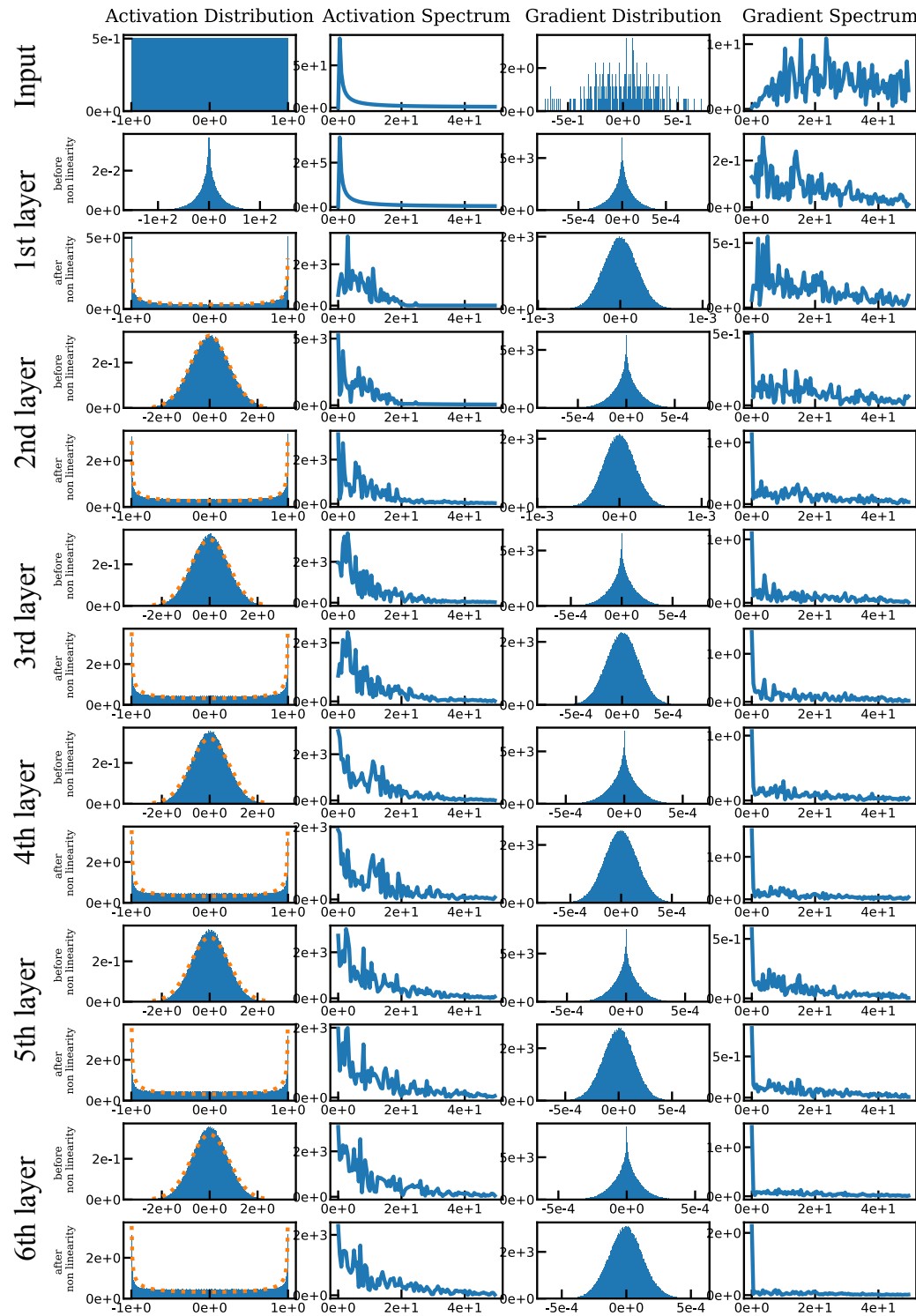

Figure 8: Activation statistics for 6 layers of a simplified sinusoidal network, demonstrating the observed distributions matched the theoretical expectation and preserves the properties from the SIREN initialization.

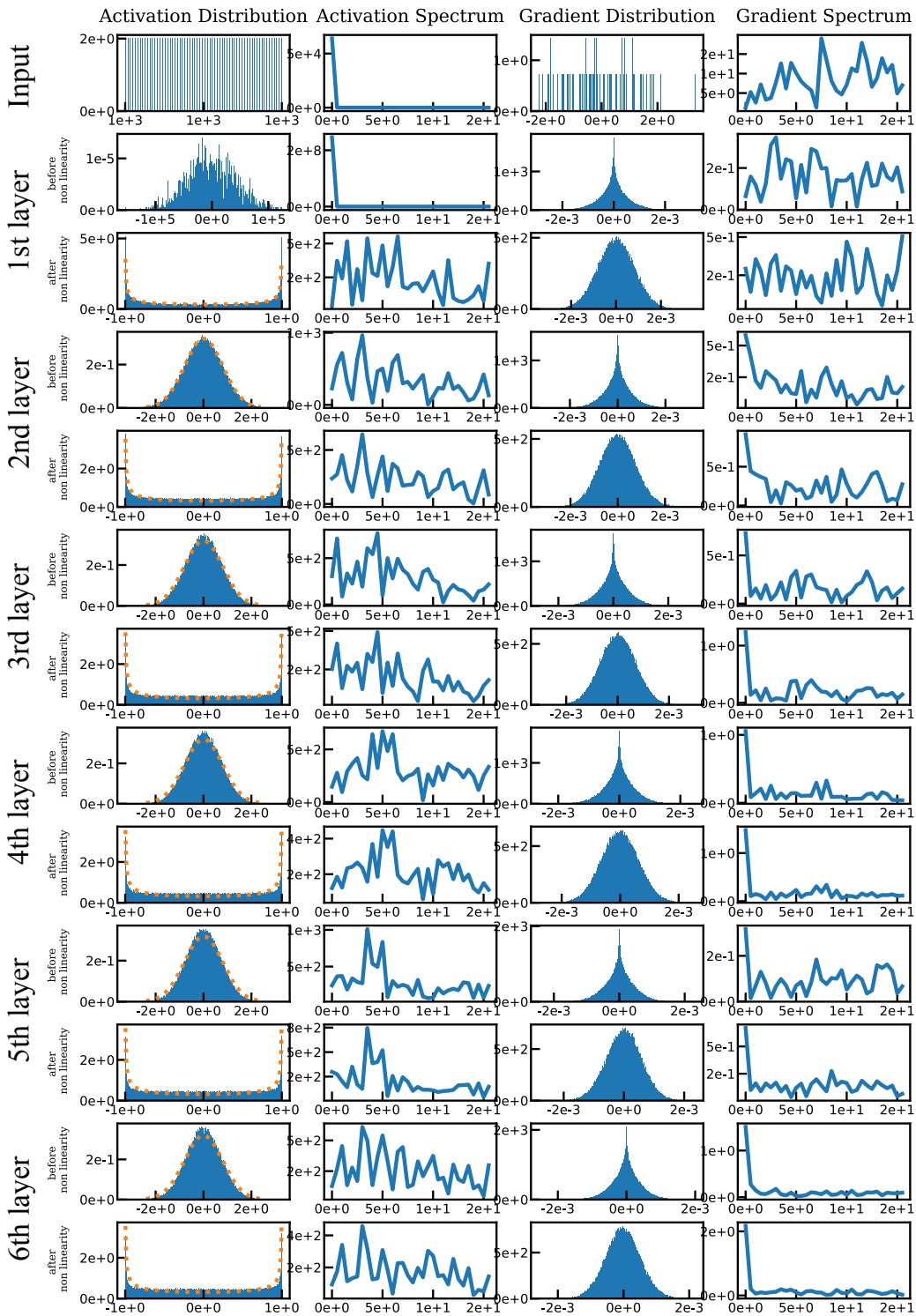

Figure 9: Activations for 6 layers of a simplified sinusoidal network in which the input has been shifted by a large value, *i.e.*, $x \rightarrow x + 1000$. The distribution characteristics are preserved, demonstrating the sinusoidal network's shift-invariance.

**Training parameters.** The input image used is $512 \times 512$, mapped to an input domain $[-1, 1]^2$. The sinusoidal network used is a 5-layer MLP with hidden size 256, following the proposed initialization scheme above. The parameter $\omega$ is set to 32. The Adam optimizer is used with a learning rate of $3 \cdot 10^{-3}$, trained for $10,000$ steps in the short duration training results and for $20,000$ steps in the long duration training results.

### C.2 POISSON

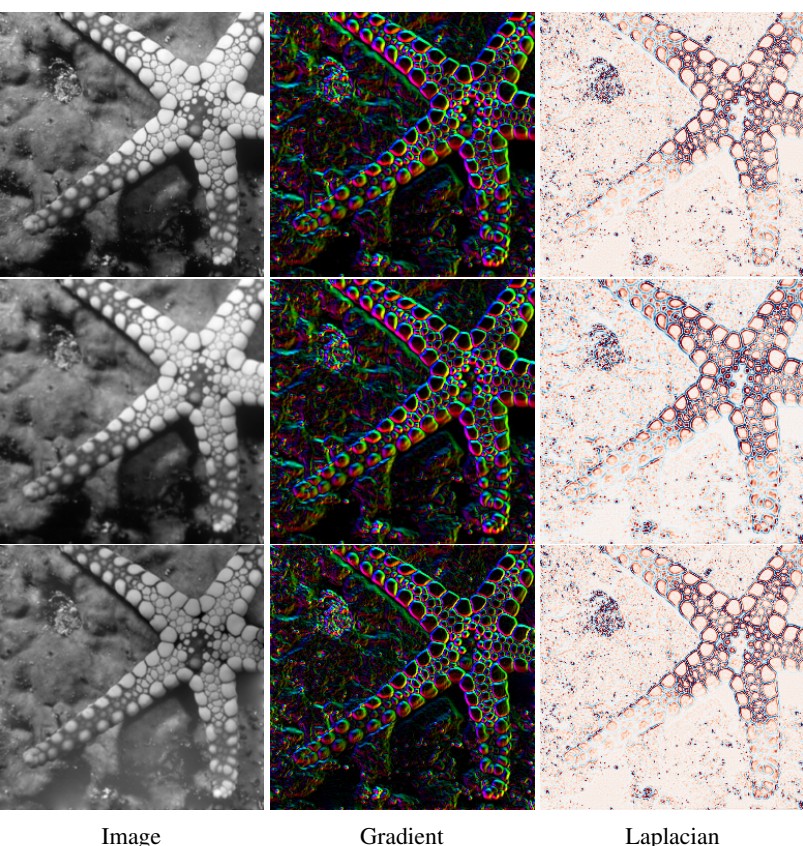

Image          Gradient          Laplacian

Figure 10: Top row: Ground truth image. Mid: Image reconstructed from sinusoidal network supervised on gradient. Bottom: Image reconstructed from sinusoidal network supervised on Laplacian.

These tasks are similar to the image fitting experiment, but instead of supervising directly on the ground truth image, the learned fitted sinusoidal network is supervised on its derivatives, constituting a Poisson problem. We perform the experiment by supervising both on the input image's gradient and Laplacian, and report the reconstruction of the image and it's gradients in each case.

Figure 10 shows the image used in this experiment, and the reconstruction from the fitted sinusoidal networks. Since reconstruction from derivatives can only be correct up to a scaling factor, we scale the reconstructions for visualization. As in the original SIREN results, we can observe that the reconstruction from the gradient is of higher quality than the one from the Laplacian.

**Training parameters.** The input image used is of size $256 \times 256$, mapped from an input domain $[-1, 1]^2$. The sinusoidal network used is a 5-layer MLP with hidden size 256, following the proposed initialization scheme above. For both experiments, the parameter $\omega$ is set to 32 and the Adam optimizer is used. For the gradient experiments, in short and long training results, a learning rate of $1 \cdot 10^{-4}$ is used, trained for $10,000$ and $20,000$ steps respectively. For the Laplace experiments, in short and long training results, a learning rate of $1 \cdot 10^{-3}$ is used, trained for $10,000$ and $20,000$ steps respectively.

## C.3 VIDEO

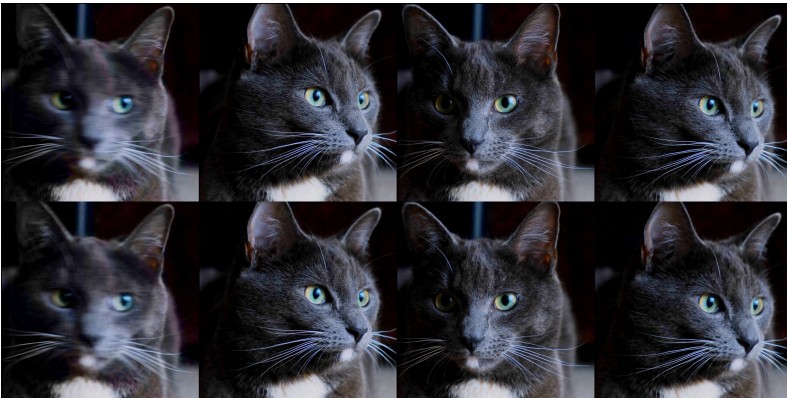

Figure 11: Top row: Frames from ground truth "cat" video. Bottom: Video reconstructed from sinusoidal network.

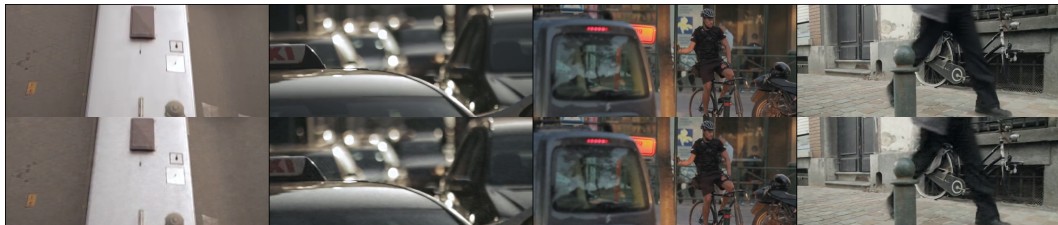

Figure 12: Top row: Frames from ground truth "bikes" video. Bottom: Video reconstructed from sinusoidal network.

These tasks are similar to the image fitting experiment, but we instead fit a video, which also has a temporal input dimension, $(t, x, y) \rightarrow (r, g, b)$. We learn a function $f : \mathbb{R}^3 \rightarrow \mathbb{R}^3$, parametrized as a sinusoidal network, in order to fit such a video.

Figures 11 and 12 show sampled frames from the videos used in this experiment, and their respective reconstructions from the fitted sinusoidal networks.

**Training parameters.** The cat video contains 300 frames of size $512 \times 512$. The bikes video contains 250 frames of size $272 \times 640$. These signals are fitted from the input domain $[-1, 1]^3$. The sinusoidal network used is a 5-layer MLP with hidden size 1024, following the proposed initialization scheme above. The parameter $\omega$ is set to 8. The Adam optimizer is used, with a learning rate of $3 \cdot 10^{-4}$ trained for $100,000$ steps in the short duration training results and for $200,000$ steps in the long duration training results.

## C.4 AUDIO

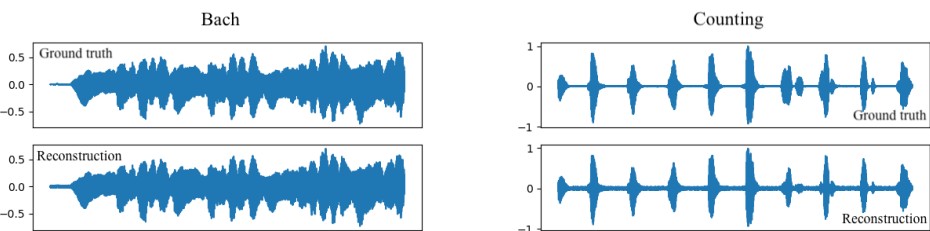

Figure 13: Ground truth and reconstructed waveforms for "Bach" and "counting" audios.

In the audio experiments, we fit an audio signal in the temporal domain as a waveform $t \to w$. We to learn a function $f : \mathbb{R} \to \mathbb{R}$, parametrized as a sinusoidal network, in order to fit the audio.

Figure 13 shows the waveforms for the input audios and the reconstructed audios from the fitted sinusoidal network.

In this experiment, we utilized a lower learning rate for the first layer compared to the rest of the network. This was used to compensate the very large $\omega$ used (in the $15,000-30,000$ range, compared to the $10-30$ range for all other experiments). One might argue that this is re-introducing complexity, counteracting the purpose the proposed simplification. However, we would claim (1) that this is only limited to cases with extremely high $\omega$, which was not present in any case except for fitting audio waves, and (2) that adjusting the learning rate for an individual layer is still an approach that is simpler and more in line with standard machine learning practice compared to multiplying all layers by a scaling factor and then adjusting their initialization variance by the same amount.

**Training parameters.** Both audios use a sampling rate of 44100Hz. The Bach audio is 7s long and the counting audio is approximately 12s long. These signals are fitted from the input domain $[-1, 1]$. The sinusoidal network used is a 5-layer MLP with hidden size 256, following the proposed initialization scheme above. For short and long training results, training is performed for $5,000$ and $50,000$ steps respectively. For the Bach experiment, the parameter $\omega$ is set to $15,000$. The Adam optimizer is used, with a general learning rate of $3 \cdot 10^{-3}$. A separate learning rate of $1 \cdot 10^{-6}$ is used for the first layer to stabilize training due to the large $\omega$ value. For the counting experiment, the parameter $\omega$ is set to $32,000$. The Adam optimizer is used, with a general learning rate of $1 \cdot 10^{-3}$ and a first layer learning rate of $1 \cdot 10^{-6}$.

## C.5 HELMHOLTZ EQUATION

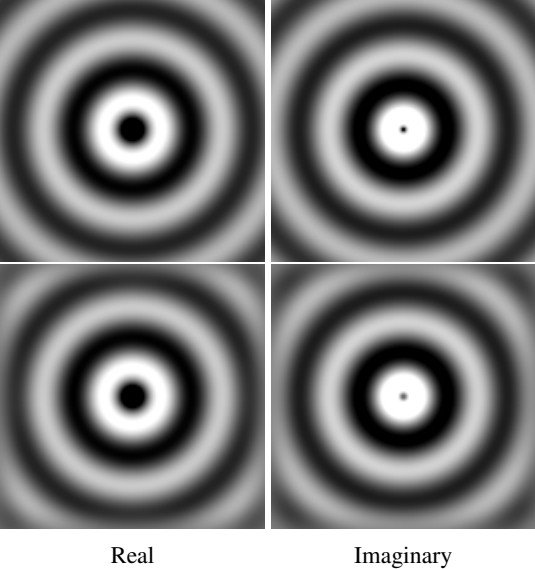

Real          Imaginary

Figure 14: Top row: Ground truth real and imaginary fields. Bottom: Reconstructed with sinusoidal network.

In this experiment we solve for the unknown wavefield $\Phi : \mathbb{R}^2 \to \mathbb{R}^2$ in the Helmholtz equation

$$(\Delta + k^2)\Phi(x) = -f(x), \tag{3}$$

with known wavenumber $k$ and source function $f$ (a Gaussian with $\mu = 0$ and $\sigma^2 = 10^{-4}$). We solve this differential equation using a sinusoidal network supervised with the physics-informed loss $\int_\Omega \|(\Delta + k^2)\Phi(x) + f(x)\|_1 dx$, evaluated at random points sampled uniformly in the domain $\Omega = [-1, 1]^2$.

Figure 14 shows the real and imaginary components of the ground truth solution to the differential equation and the solution recovered by the fitted sinusoidal network.

**Training parameters.** The sinusoidal network used is a 5-layer MLP with hidden size 256, following the proposed initialization scheme above. The parameter $\omega$ is set to 16. The Adam optimizer is used, with a learning rate of $3 \cdot 10^{-4}$ trained for $50,000$ steps.

### C.6 SIGNED DISTANCE FUNCTION (SDF)

In these tasks we learn a 3D signed distance function. We learn a function $f : \mathbb{R}^3 \to \mathbb{R}$, parametrized as a sinusoidal network, to model a signed distance function representing a 3D scene. This function is supervised indirectly from point cloud data of the scene. Figures 16 and 15 show 3D renderings of the volumes inferred from the learned SDFs.

**Training parameters.** The statue point cloud contains $4,999,996$ points. The room point cloud contains $10,250,688$ points. These signals are fitted from the input domain $[-1,1]^3$. The sinusoidal network used is a 5-layer MLP with hidden size 256 for the statue and 1024 for the room. The parameter $\omega$ is set to 4. The Adam optimizer is used, with a learning rate of $8 \cdot 10^{-4}$ and a batch size of 1400. All models are trained for $190,000$ steps for the statue experiment and for $410,000$ steps for the room experiment.

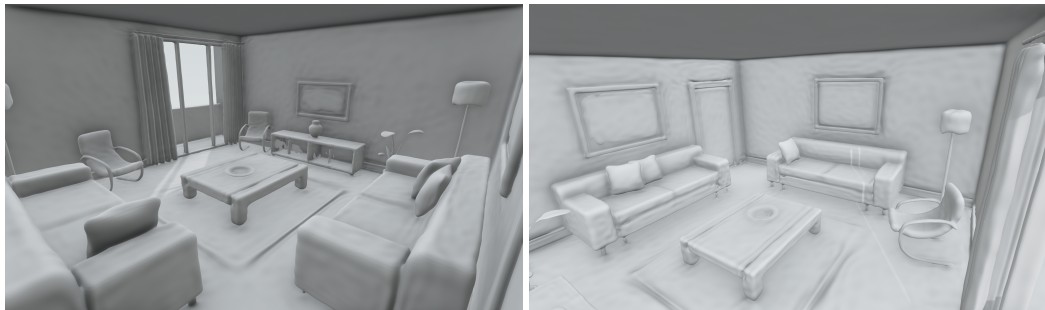

Figure 15: Rendering of the "room" 3D scene SDF learned by the sinusoidal network from a point cloud.

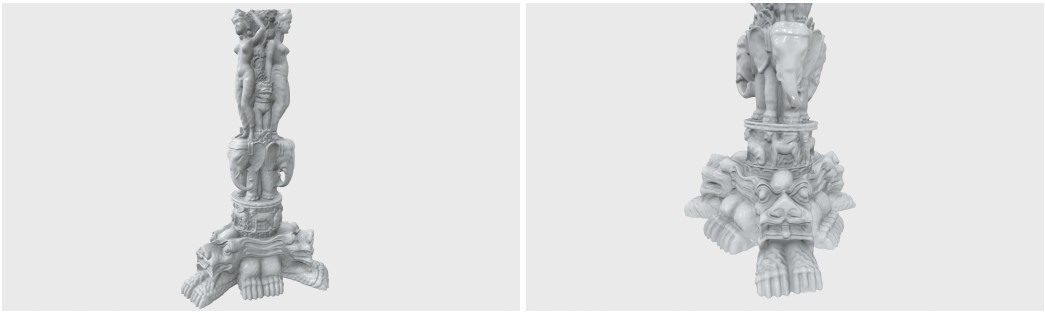

Figure 16: Rendering of the "statue" 3D scene SDF learned by the sinusoidal network from a point cloud.

## D NEURAL TANGENT KERNEL ANALYSIS AND PROOFS

### D.1 PRELIMINARIES

In order to perform the subsequent NTK analysis, we first need to formalize definitions for simple sinusoidal networks and SIRENs. The definitions used here adhere to the common NTK analysis practices, and thus differ slightly from practical implementation.

**Definition 1.** *For the purposes of the following proofs, a (sinusoidal) fully-connected neural network with $L$ hidden layers that takes as input $x \in \mathbb{R}^{n_0}$, is defined as the function $f^{(L)} : \mathbb{R}^{n_0} \to \mathbb{R}^{n_{L+1}}$, recursively given by*

$$f^{(0)}(x) = \omega \left( W^{(0)}x + b^{(0)} \right),$$

$$f^{(L)}(x) = W^{(L)} \frac{1}{\sqrt{n_L}} \sin\left( f^{(L-1)} \right) + b^{(L)},$$

*where $\omega \in \mathbb{R}$. The parameters $\left\{ W^{(j)} \right\}_{j=0}^L$ have shape $n_{j+1} \times n_j$ and all have each element sampled independently either from $\mathcal{N}(0,1)$ (for simple sinusoidal networks) or from $\mathcal{U}(-c,c)$ with some bound $c \in \mathbb{R}$ (for SIRENs). The $\left\{ b^{(j)} \right\}_{j=0}^L$ are $n_{j+1}$-dimensional vectors sampled independently from $\mathcal{N}(0, I_{n_{j+1}})$.*

With this definition, we now state the general formulation of the NTK, which applies in general to fully-connected networks with Lipschitz non-linearities, and consequently in particular to the sinusoidal networks studied here as well. Let us first define the NNGP, which has covariance recursively defined by

$$\Sigma^{(L+1)}(x, \tilde{x}) = \mathbb{E}_{f \sim \mathcal{N}(0, \Sigma^{(L)})} \left[ \sigma(f(x))\sigma(f(\tilde{x})) \right] + \beta^2,$$

with base case $\Sigma^{(1)}(x, \tilde{x}) = \frac{1}{n_0} x^T \tilde{x} + \beta^2$, and where $\beta$ gives the variance of the bias terms in the neural network layers (Neal, 1994; Lee et al., 2018). Now the NTK is given by the following theorem.

**Theorem 6.** *For a neural network with $L$ hidden layers $f^{(L)} : \mathbb{R}^{n_0} \to \mathbb{R}^{n_{L+1}}$ following Definition 1, as the size of the hidden layers $n_1, \ldots, n_L \to \infty$ sequentially, the neural tangent kernel (NTK) of $f^{(L)}$ converges in probability to the deterministic kernel $\Theta^{(L)}$ defined recursively as*

$$\Theta^{(0)}(x, \tilde{x}) = \Sigma^{(0)}(x, \tilde{x}) = \omega^2 \left( x^T \tilde{x} + 1 \right),$$

$$\Theta^{(L)}(x, \tilde{x}) = \Theta^{(L-1)}(x, \tilde{x}) \dot{\Sigma}^{(L)}(x, \tilde{x}) + \Sigma^{(L)}(x, \tilde{x}),$$

*where $\left\{ \Sigma^{(l)} \right\}_{l=0}^L$ are the neural network Gaussian processes (NNGPs) corresponding to each $f^{(l)}$ and*

$$\dot{\Sigma}^{(l)}(x, \tilde{x}) = \mathbb{E}_{(u,v) \sim \Sigma^{(l-1)}(x, \tilde{x})} \left[ \cos(u)\cos(v) \right].$$

*Proof.* This is a standard general NTK theorem, showing that the limiting kernel recursively in terms of the network's NNGPs and the previous layer's NTK. For brevity we omit the proof here and refer the reader to, for example, Jacot et al. (2020).

The only difference is for the base case $\Sigma^{(0)}$, due to the fact that we have an additional $\omega$ parameter in the first layer. It is simple to see that the neural network with 0 hidden layers, *i.e.* the linear model $\omega \left( W^{(0)}x + b^{(0)} \right)$ will lead to the same Gaussian process covariance kernel as the original proof, $x^T \tilde{x} + 1$, only adjusted by the additional variance factor $\omega^2$. □

Theorem 6 demonstrates that the NTK can be constructed as a recursive function of the NTK of previous layers and the network's NNGPs. In the following sections we will derive the NNGPs for the SIREN and the simple sinusoidal network directly. We will then use these NNGPs with Theorem 6 to derive their NTKs as well.

To finalize this preliminary section, we also provide two propositions that will be useful in following proofs in this section.

**Proposition 7.** *For any $\omega \in \mathbb{R}$, $x \in \mathbb{R}^d$,*

$$\mathbb{E}_{w \sim \mathcal{N}(0, I_d)} \left[ e^{i\omega \left( w^T x \right)} \right] = e^{-\frac{\omega^2}{2} \|x\|_2^2}$$

*Proof.* Omitting $w \sim \mathcal{N}(0, I_d)$ from the expectation for brevity, we have

$$\mathbb{E}\left[e^{i\omega\left(w^T x\right)}\right] = \mathbb{E}\left[e^{i\omega \sum_{j=1}^{d} w_j x_j}\right].$$

By independence of the components of $w$ and the definition of expectation,

$$\mathbb{E}\left[e^{i\omega \sum_{j=1}^{d} iw_j x_j}\right] = \prod_{j=1}^{d} \mathbb{E}\left[e^{i\omega \, w_j x_j}\right] = \prod_{j=1}^{d} \frac{1}{\sqrt{2\pi}} \int_{-\infty}^{\infty} e^{i\omega \, w_j x_j} e^{-\frac{w_j^2}{2}} dw_j.$$

Completing the square, we get

$$\prod_{j=1}^{d} \frac{1}{\sqrt{2\pi}} \int_{-\infty}^{\infty} e^{i\omega \, w_j x_j} e^{-\frac{1}{2} w_j^2} dw_j = \prod_{j=1}^{d} \frac{1}{\sqrt{2\pi}} \int_{-\infty}^{\infty} e^{\frac{1}{2}\left(i^2 \omega^2 x_j^2 - i^2 \omega^2 x_j^2 + 2ix_j w_j - w_j^2\right)} dw_j$$

$$= \prod_{j=1}^{d} e^{\frac{1}{2}i^2 \omega^2 x_j^2} \frac{1}{\sqrt{2\pi}} \int_{-\infty}^{\infty} e^{-\frac{1}{2}\left(i^2 \omega^2 x_j^2 - 2i\omega^2 x_j w_j + w_j^2\right)} dw_j$$

$$= \prod_{j=1}^{d} e^{-\frac{1}{2}\omega^2 x_j^2} \frac{1}{\sqrt{2\pi}} \int_{-\infty}^{\infty} e^{-\frac{1}{2}\left(w_j - i\omega x_j\right)^2} dw_j.$$

Since the integral and its preceding factor constitute a Gaussian pdf, they integrate to 1, leaving the final result

$$\prod_{j=1}^{d} e^{-\frac{\omega^2}{2} x_j^2} = e^{-\frac{\omega^2}{2} \sum_{j=1}^{d} x_j^2} = e^{-\frac{\omega^2}{2} \|x_j\|_2^2}.$$

$\square$

**Proposition 8.** *For any $c, \omega \in \mathbb{R}$, $x \in \mathbb{R}^d$,*

$$\mathbb{E}_{w \sim \mathcal{U}_d(-c,c)}\left[e^{i\omega\left(w^T x\right)}\right] = \prod_{j=1}^{d} \operatorname{sinc}(c\,\omega x_j).$$

*Proof.* Omitting $w \sim \mathcal{U}_d(-c, c)$ from the expectation for brevity, we have

$$\mathbb{E}\left[e^{i\omega\left(w^T x\right)}\right] = \mathbb{E}\left[e^{i\omega \sum_{j=1}^{d} w_j x_j}\right].$$

By independence of the components of $w$ and the definition of expectation,

$$\mathbb{E}\left[e^{i\omega \sum_{j=1}^{d} w_j x_j}\right] = \prod_{j=1}^{d} \mathbb{E}\left[e^{i\omega \, w_j x_j}\right] = \prod_{j=1}^{d} \int_{-c}^{c} e^{i\omega \, w_j x_j} \frac{1}{2c} dw_j = \prod_{j=1}^{d} \frac{1}{2c} \int_{-c}^{c} e^{i\omega \, w_j x_j} dw_j.$$

Now, focusing on the integral above, we have

$$\int_{-c}^{c} e^{i\omega \, w_j x_j} dw_j = \int_{-c}^{c} \cos(\omega \, w_j x_j) dw_j + i \int_{-c}^{c} \sin(\omega \, w_j x_j) dw_j$$

$$= \left. \frac{\sin(\omega \, w_j x_j)}{\omega x_j} \right|_{-c}^{c} - i \left. \frac{\cos(\omega \, w_j x_j)}{\omega x_j} \right|_{-c}^{c}$$

$$= \frac{2\sin(c\,\omega x_j)}{\omega x_j}.$$

Finally, plugging this back into the product above, we get

$$\prod_{j=1}^{d} \frac{1}{2c} \int_{-c}^{c} e^{i\omega \, w_j x_j} dw_j = \prod_{j=1}^{d} \frac{1}{2c} \frac{2\sin(c\,\omega x_j)}{\omega x_j} = \prod_{j=1}^{d} \operatorname{sinc}(c\,\omega x_j).$$

$\square$

## D.2 Shallow sinusoidal networks

For the next few proofs, we will be focusing on neural networks with a single hidden layer, *i.e.* $L = 1$. Expanding the definition above, such a network is given by

$$f^{(1)}(x) = W^{(1)} \frac{1}{\sqrt{n_1}} \sin\left(\omega\left(W^{(0)}x + b^{(0)}\right)\right) + b^{(1)}. \tag{4}$$

The advantage of analysing such shallow networks is that their NNGPs and NTKs have formulations that are intuitively interpretable, providing insight into their characteristics. We later extend these derivations to networks of arbitrary depth.

### D.2.1 SIREN

First, let us derive the NNGP for a SIREN with a single hidden layer.

**Theorem 9.** *Shallow SIREN NNGP. For a single hidden layer SIREN $f^{(1)} : \mathbb{R}^{n_0} \to \mathbb{R}^{n_2}$ following Definition 1, as the size of the hidden layer $n_1 \to \infty$, $f^{(1)}$ tends (by law of large numbers) to the neural network Gaussian Process (NNGP) with covariance*

$$\Sigma^{(1)}(x, \tilde{x}) = \frac{c^2}{6} \left[ \prod_{j=1}^{n_0} \operatorname{sinc}(c\,\omega\,(x_j - \tilde{x}_j)) - e^{-2\omega^2} \prod_{j=1}^{n_0} \operatorname{sinc}(c\,\omega\,(x_j + \tilde{x}_j)) \right] + 1.$$

*Proof.* We first show that despite the usage of a uniform distribution for the weights, this initialization scheme still leads to an NNGP. In this initial part, we follow an approach similar to Lee et al. (2018), with the modifications necessary for this conclusion to hold.

From our neural network definition, each element $f^{(1)}(x)_j$ in the output vector is a weighted combination of elements in $W^{(1)}$ and $b^{(1)}$. Conditioning on the outputs from the first layer ($L = 0$), since the sine function is bounded and each of the parameters is uniformly distributed with finite variance and zero mean, the $f^{(1)}(x)_j$ become normally distributed with mean zero as $n_1 \to \infty$ by the (Lyapunov) central limit theorem (CLT). Since any subset of elements in $f^{(1)}(x)$ is jointly Gaussian, we have that this outer layer is described by a Gaussian process.

Now that we have concluded that this initialization scheme still entails an NNGP, we have that its covariance is determined by $\sigma_W^2 \Sigma^{(1)} + \sigma_b^2 = \frac{c^2}{3} \Sigma^{(1)} + 1$, where

$$\Sigma^{(1)}(x, \tilde{x}) = \lim_{n_1 \to \infty} \left[ \frac{1}{n_1} \left\langle \sin\left(f^{(0)}(x)\right), \sin\left(f^{(0)}(\tilde{x})\right) \right\rangle \right]$$

$$= \lim_{n_1 \to \infty} \left[ \frac{1}{n_1} \sum_{j=1}^{n_1} \sin\left(f^{(0)}(x)\right)_j \sin\left(f^{(0)}(\tilde{x})\right)_j \right]$$

$$= \lim_{n_1 \to \infty} \left[ \frac{1}{n_1} \sum_{j=1}^{n_1} \sin\left(\omega\left(W_j^{(0)}x + b_j^{(0)}\right)\right) \sin\left(\omega\left(W_j^{(0)}\tilde{x} + b_j^{(0)}\right)\right) \right].$$

Now by the law of large number (LLN) the limit above converges to

$$\mathbb{E}_{w \sim \mathcal{U}_{n_0}(-c,c),\, b \sim \mathcal{N}(0,1)} \left[ \sin\left(\omega\left(w^T x + b\right)\right) \sin\left(\omega\left(w^T \tilde{x} + b\right)\right) \right],$$

where $w \in \mathbb{R}^{n_0}$ and $b \in \mathbb{R}$. Omitting the distributions from the expectation for brevity and expanding the exponential definition of sine, we have

$$\mathbb{E}\left[ \frac{1}{2i} \left( e^{i\omega\left(w^T x + b\right)} - e^{-i\omega\left(w^T x + b\right)} \right) \frac{1}{2i} \left( e^{i\omega\left(w^T \tilde{x} + b\right)} - e^{-i\omega\left(w^T \tilde{x} + b\right)} \right) \right]$$

$$= -\frac{1}{4} \mathbb{E}\left[ e^{i\omega\left(w^T x + b\right) + i\omega\left(w^T \tilde{x} + b\right)} - e^{i\omega\left(w^T x + b\right) - i\omega\left(w^T \tilde{x} + b\right)} - e^{-i\omega\left(w^T x + b\right) + i\omega\left(w^T \tilde{x} + b\right)} + e^{-i\omega\left(w^T x + b\right) - i\omega\left(w^T \tilde{x} + b\right)} \right]$$

$$= -\frac{1}{4} \left[ \mathbb{E}\left[ e^{i\omega\left(w^T (x+\tilde{x})\right)} \right] \mathbb{E}\left[ e^{2i\omega b} \right] - \mathbb{E}\left[ e^{i\omega\left(w^T (x-\tilde{x})\right)} \right] - \mathbb{E}\left[ e^{i\omega\left(w^T (\tilde{x}-x)\right)} \right] + \mathbb{E}\left[ e^{i\omega\left(w^T (-x-\tilde{x})\right)} \right] \mathbb{E}\left[ e^{-2i\omega b} \right] \right]$$

Applying Propositions 7 and 8 to each expectation above and noting that the sinc function is even, we are left with

$$-\frac{1}{4}\left[2\prod_{j=1}^{n_0}\mathrm{sinc}(c\,\omega\,(x_j+\tilde{x}_j)) - 2e^{-2\omega^2}\prod_{j=1}^{n_0}\mathrm{sinc}(c\,\omega\,(x_j-\tilde{x}_j))\right]$$

$$=\frac{1}{2}\left[\prod_{j=1}^{n_0}\mathrm{sinc}(c\,\omega\,(x_j-\tilde{x}_j)) - e^{-2\omega^2}\prod_{j=1}^{n_0}\mathrm{sinc}(c\,\omega\,(x_j+\tilde{x}_j))\right].$$

$\square$

For simplicity, if we take the case of a one-dimensional output (*e.g.*, an audio signal or a monochromatic image) with the standard SIREN setting of $c=\sqrt{6}$, the NNGP reduces to

$$\Sigma^{(1)}(x,\tilde{x}) = \mathrm{sinc}\left(\sqrt{6}\,\omega\,(x-\tilde{x})\right) - e^{-2\omega^2}\mathrm{sinc}\left(\sqrt{6}\,\omega\,(x+\tilde{x})\right) + 1.$$

We can already notice that this kernel is composed of sinc functions. The sinc function is the ideal low-pass filter. For any value of $\omega > 1$, we can see the the first term in the expression above will completely dominate the expression, due to the exponential $e^{-2\omega^2}$ factor. In practice, $\omega$ is commonly set to values at least one order of magnitude above 1, if not multiple orders of magnitude above that in certain cases (*e.g.*, high frequency audio signals). This leaves us with simply

$$\Sigma^{(1)}(x,\tilde{x}) = \mathrm{sinc}\left(\sqrt{6}\,\omega\,(x-\tilde{x})\right) + 1.$$

Notice that not only does our kernel reduce to the sinc function, but it also reduces to a function solely of $\Delta x = x - \tilde{x}$. This agrees with the shift-invariant property we observe in SIRENs, since the NNGP is dependent only on $\Delta x$, but not on the particular values of $x$ and $\tilde{x}$. Notice also that $\omega$ defines the bandwidth of the sinc function, thus determining the maximum frequencies it allows to pass.

The general sinc form and the shift-invariance of this kernel can be visualized in Figure 17, along with the effect of varying $\omega$ on the bandwidth of the NNGP kernel.

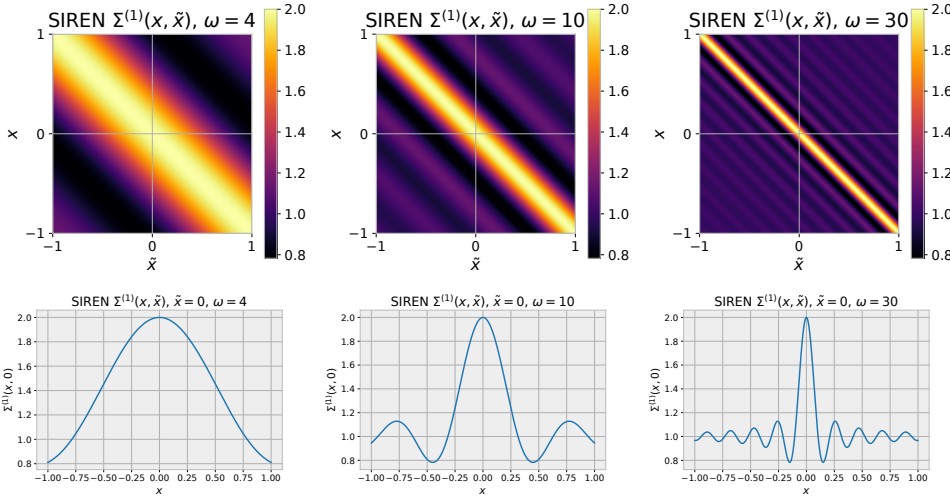

Figure 17: The NNGP for SIREN at different $\omega$ values. The top row shows the kernel values for pairs $(x,\tilde{x}) \in [-1,1]^2$. Bottom row shows a slice at fixed $\tilde{x}=0$.

We can see that the NTK of the shallow SIREN, derived below, maintains the same relevant characteristics as the NNGP. We first derive $\dot{\Sigma}$ in the Lemma below.

**Lemma 10.** *For $\omega \in \mathbb{R}$, $\dot{\Sigma}^{(1)}(x, \tilde{x}) : \mathbb{R}^{n_0} \times \mathbb{R}^{n_0} \to \mathbb{R}$ is given by*

$$\Sigma^{(1)}(x, \tilde{x}) = \frac{c^2}{6} \left[ \prod_{j=1}^{n_0} \text{sinc}(c\,\omega\,(x_j - \tilde{x}_j)) + e^{-2\omega^2} \prod_{j=1}^{n_0} \text{sinc}(c\,\omega\,(x_j + \tilde{x}_j)) \right] + 1.$$

*Proof.* The proof follows the same pattern as Theorem 9, with the only difference being a few sign changes after the exponential expansion of the trigonometric functions, due to the different identities for sine and cosine. □

Now we can derive the NTK for the shallow SIREN.

**Corollary 11.** *Shallow SIREN NTK. For a single hidden layer SIREN $f^{(1)} : \mathbb{R}^{n_0} \to \mathbb{R}^{n_2}$ following Definition 1, its neural tangent kernel (NTK), as defined in Theorem 6, is given by*

$$\Theta^{(1)}(x, \tilde{x}) = \left( \omega^2 \left( x^T \tilde{x} + 1 \right) \right) \left( \frac{c^2}{6} \left[ \prod_{j=1}^{n_0} \text{sinc}(c\,\omega\,(x_j - \tilde{x}_j)) - e^{-2\omega^2} \prod_{j=1}^{n_0} \text{sinc}(c\,\omega\,(x_j + \tilde{x}_j)) \right] + 1 \right)$$

$$+ \frac{c^2}{6} \left[ \prod_{j=1}^{n_0} \text{sinc}(c\,\omega\,(x_j - \tilde{x}_j)) + e^{-2\omega^2} \prod_{j=1}^{n_0} \text{sinc}(c\,\omega\,(x_j + \tilde{x}_j)) \right] + 1$$

$$= \frac{c^2}{6} \left( \omega^2 \left( x^T \tilde{x} + 1 \right) + 1 \right) \prod_{j=1}^{n_0} \text{sinc}(c\,\omega\,(x_j - \tilde{x}_j))$$

$$- \frac{c^2}{6} \left( \omega^2 \left( x^T \tilde{x} + 1 \right) - 1 \right) e^{-2\omega^2} \prod_{j=1}^{n_0} \text{sinc}(c\,\omega\,(x_j + \tilde{x}_j)) + \omega^2 \left( x^T \tilde{x} + 1 \right) + 1.$$

*Proof.* Follows trivially by applying Theorem 9 and Lemma 10 to Theorem 6. □

Though the expressions become more complex due to the formulation of the NTK, we can see that many of the same properties from the NNGP still apply. Again, for reasonable values of $\omega$, the term with the exponential factor $e^{-2\omega^2}$ will be of negligible relative magnitude. With $c = \sqrt{6}$, this leaves us with

$$\left( \omega^2 \left( x^T \tilde{x} + 1 \right) + 1 \right) \prod_{j=1}^{n_0} \text{sinc}\left( \sqrt{6}\,\omega\,(x_j - \tilde{x}_j) \right) + \omega^2 \left( x^T \tilde{x} + 1 \right) + 1,$$

which is of the same form as the NNGP, with some additional linear terms $x^T \tilde{x}$. Though these linear terms break the pure shift-invariance, we still have a strong diagonal and the sinc form with bandwidth determined by $\omega$, as can be seen in Figure 18.

Similarly to the NNGP, the SIREN NTK suggests that training a shallow SIREN is approximately equivalent to performing kernel regression with a sinc kernel, a low-pass filter, with its bandwidth defined by $\omega$. This agrees intuitively with the experimental observations from the paper that in order to fit higher frequencies signals, a larger $\omega$ is required.

### D.2.2 SIMPLE SINUSOIDAL NETWORK

Just as we did in the last section, we will now first derive the NNGP for a simple sinusoidal network, and then use that in order to obtain its NTK as well. As we will see, the Gaussian initialization employed in the SSN has the benefit of rendering the derivations cleaner, while retaining the relevant properties from the SIREN initialization. We observe that a similar derivation of this NNGP (using cosine functions instead of sine) can be found in Pearce et al. (2019), with a focus on a Bayesian perspective for the result.

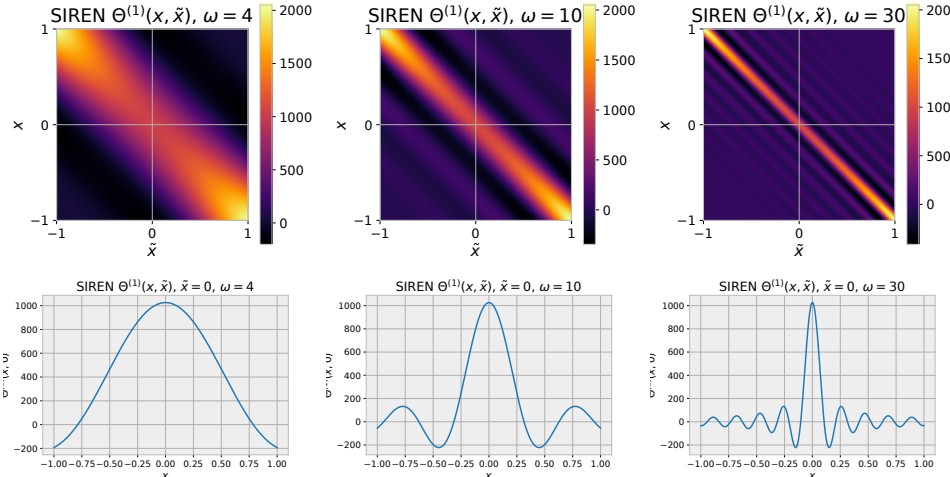

Figure 18: The NTK for SIREN at different $\omega$ values. The top row shows the kernel values for pairs $(x, \tilde{x}) \in [-1, 1]^2$. Bottom row shows a slice at fixed $\tilde{x} = 0$.

**Theorem 12.** *Shallow SSN NNGP. For a single hidden layer simple sinusoidal network $f^{(1)}$ : $\mathbb{R}^{n_0} \to \mathbb{R}^{n_2}$ following Definition 1, as the size of the hidden layer $n_1 \to \infty$, $f^{(1)}$ tends (by law of large numbers) to the neural network Gaussian Process (NNGP) with covariance*

$$\Sigma^{(1)}(x, \tilde{x}) = \frac{1}{2} \left( e^{-\frac{\omega^2}{2} \|x - \tilde{x}\|_2^2} - e^{-\frac{\omega^2}{2} \|x + \tilde{x}\|_2^2} e^{-2\omega^2} \right) + 1.$$

*Proof.* We again initially follow an approach similar to the one described in Lee et al. (2018).

From our sinusoidal network definition, each element $f^{(1)}(x)_j$ in the output vector is a weighted combination of elements in $W^{(1)}$ and $b^{(1)}$. Conditioning on the outputs from the first layer ($L = 0$), since the sine function is bounded and each of the parameters is Gaussian with finite variance and zero mean, the $f^{(1)}(x)_j$ are also normally distributed with mean zero by the CLT. Since any subset of elements in $f^{(1)}(x)$ is jointly Gaussian, we have that this outer layer is described by a Gaussian process.

Therefore, its covariance is determined by $\sigma_W^2 \Sigma^{(1)} + \sigma_b^2 = \Sigma^{(1)} + 1$, where

$$
\begin{aligned}
\Sigma^{(1)}(x, \tilde{x}) &= \lim_{n_1 \to \infty} \left[ \frac{1}{n_1} \left\langle \sin\left(f^{(0)}(x)\right), \sin\left(f^{(0)}(\tilde{x})\right) \right\rangle \right] \\
&= \lim_{n_1 \to \infty} \left[ \frac{1}{n_1} \sum_{j=1}^{n_1} \sin\left(f^{(0)}(x)\right)_j \sin\left(f^{(0)}(\tilde{x})\right)_j \right] \\
&= \lim_{n_1 \to \infty} \left[ \frac{1}{n_1} \sum_{j=1}^{n_1} \sin\left(\omega \left(W_j^{(0)} x + b_j^{(0)}\right)\right) \sin\left(\omega \left(W_j^{(0)} \tilde{x} + b_j^{(0)}\right)\right) \right].
\end{aligned}
$$

Now by the LLN the limit above converges to

$$\mathbb{E}_{w \sim \mathcal{N}(0, I_{n_0}), b \sim \mathcal{N}(0,1)} \left[ \sin\left(\omega \left(w^T x + b\right)\right) \sin\left(\omega \left(w^T \tilde{x} + b\right)\right) \right],$$

where $w \in \mathbb{R}^{n_0}$ and $b \in \mathbb{R}$. Omitting the distributions from the expectation for brevity and expanding the exponential definition of sine, we have

$$\mathbb{E}\left[\frac{1}{2i}\left(e^{i\omega(w^T x+b)} - e^{-i\omega(w^T x+b)}\right)\frac{1}{2i}\left(e^{i\omega(w^T \tilde{x}+b)} - e^{-i\omega(w^T \tilde{x}+b)}\right)\right]$$

$$= -\frac{1}{4}\mathbb{E}\left[e^{i\omega(w^T x+b)+i\omega(w^T \tilde{x}+b)} - e^{i\omega(w^T x+b)-i\omega(w^T \tilde{x}+b)} - e^{-i\omega(w^T x+b)+i\omega(w^T \tilde{x}+b)} + e^{-i\omega(w^T x+b)-i\omega(w^T \tilde{x}+b)}\right]$$

$$= -\frac{1}{4}\left[\mathbb{E}\left[e^{i\omega(w^T(x+\tilde{x}))}\right]\mathbb{E}\left[e^{2i\omega b}\right] - \mathbb{E}\left[e^{i\omega(w^T(x-\tilde{x}))}\right] - \mathbb{E}\left[e^{i\omega(w^T(\tilde{x}-x))}\right] + \mathbb{E}\left[e^{i\omega(w^T(-x-\tilde{x}))}\right]\mathbb{E}\left[e^{-2i\omega b}\right]\right]$$

Applying Proposition 7 to each expectation above, it becomes

$$-\frac{1}{4}\left(e^{-\frac{\omega^2}{2}\|x+\tilde{x}\|_2^2}e^{-2\omega^2} - e^{-\frac{\omega^2}{2}\|x-\tilde{x}\|_2^2} - e^{-\frac{\omega^2}{2}\|x+\tilde{x}\|_2^2} + e^{-\frac{\omega^2}{2}\|x+\tilde{x}\|_2^2}e^{-2\omega^2}\right)$$

$$= \frac{1}{2}\left(e^{-\frac{\omega^2}{2}\|x-\tilde{x}\|_2^2} - e^{-\frac{\omega^2}{2}\|x+\tilde{x}\|_2^2}e^{-2\omega^2}\right).$$

$\square$

We an once again observe that, for practical values of $\omega$, the NNGP simplifies to

$$\frac{1}{2}e^{-\frac{\omega^2}{2}\|x-\tilde{x}\|_2^2} + 1.$$

This takes the form of a Gaussian kernel, which is also a low-pass filter, with its bandwidth determined by $\omega$. We note that, similar to the $c = \sqrt{6}$ setting from SIRENs, in practice a scaling factor of $\sqrt{2}$ is applied to the normal activations, as described in Section 3, which cancels out the $1/2$ factors from the kernels, preserving the variance magnitude.

Moreover, we can also observe again that the kernel is a function solely of $\Delta x$, in agreement with the shift invariance that is also observed in simple sinusoidal networks. Visualizations of this NNGP are provided in Figure 19.

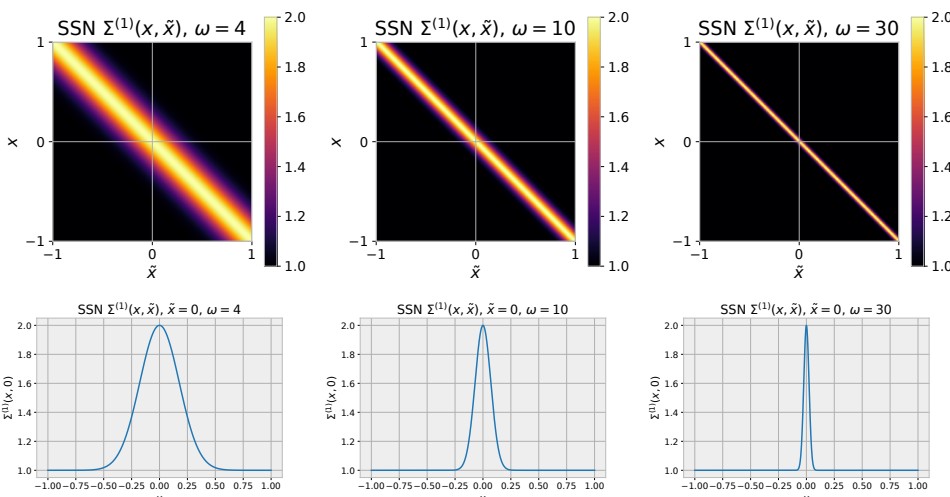

Figure 19: The NNGP for SSN at different $\omega$ values. The top row shows the kernel values for pairs $(x, \tilde{x}) \in [-1, 1]^2$. Bottom row shows a slice at fixed $\tilde{x} = 0$.

We will now proceed to derive the NTK, which requires first obtaining $\dot{\Sigma}$.

**Lemma 13.** *For $\omega \in \mathbb{R}$, $\dot{\Sigma}^{(1)}(x, \tilde{x}) : \mathbb{R}^{n_0} \times \mathbb{R}^{n_0} \to \mathbb{R}$ is given by*

$$\dot{\Sigma}^{(1)}(x, \tilde{x}) = \frac{1}{2}\left(e^{-\frac{\omega^2}{2}\|x-\tilde{x}\|_2^2} + e^{-\frac{\omega^2}{2}\|x+\tilde{x}\|_2^2}e^{-2\omega^2}\right) + 1.$$

*Proof.* The proof follows the same pattern as Theorem 12, with the only difference being a few sign changes after the exponential expansion of the trigonometric functions, due to the different identities for sine and cosine.

$\square$

**Corollary 14.** *Shallow SSN NTK. For a simple sinusoidal network with a single hidden layer* $f^{(1)} : \mathbb{R}^{n_0} \to \mathbb{R}^{n_2}$ *following Definition 1, its neural tangent kernel (NTK), as defined in Theorem 6, is given by*

$$
\Theta^{(1)}(x, \tilde{x}) = \left( \omega^2 \left( x^T \tilde{x} + 1 \right) \right) \left[ \frac{1}{2} \left( e^{-\frac{\omega^2}{2} \|x - \tilde{x}\|_2^2} + e^{-\frac{\omega^2}{2} \|x + \tilde{x}\|_2^2} e^{-2\omega^2} \right) + 1 \right]
$$
$$
+ \frac{1}{2} \left( e^{-\frac{\omega^2}{2} \|x - \tilde{x}\|_2^2} - e^{-\frac{\omega^2}{2} \|x + \tilde{x}\|_2^2} e^{-2\omega^2} \right) + 1
$$
$$
= \frac{1}{2} \left( \omega^2 \left( x^T \tilde{x} + 1 \right) + 1 \right) e^{-\frac{\omega^2}{2} \|x - \tilde{x}\|_2^2}
$$
$$
- \frac{1}{2} \left( \omega^2 \left( x^T \tilde{x} + 1 \right) - 1 \right) e^{-\frac{\omega^2}{2} \|x + \tilde{x}\|_2^2} e^{-2\omega^2} + \omega^2 \left( x^T \tilde{x} + 1 \right) + 1.
$$

*Proof.* Follows trivially by applying Theorem 12 and Lemma 13 to Theorem 6. $\square$

We again note the vanishing factor $e^{-2\omega^2}$, which leaves us with

$$
\frac{1}{2} \left( \omega^2 \left( x^T \tilde{x} + 1 \right) + 1 \right) e^{-\frac{\omega^2}{2} \|x - \tilde{x}\|_2^2} + \omega^2 \left( x^T \tilde{x} + 1 \right) + 1. \tag{5}
$$

As with the SIREN before, this NTK is still of the same form as its corresponding NNGP. While again we have additional linear terms $x^T \tilde{x}$ in the NTK compared to the NNGP, in this case as well the kernel preserves its strong diagonal. It is still close to a Gaussian kernel, with its bandwidth determined directly by $\omega$. We demonstrate this in Figure 20, where the NTK for different values of $\omega$ is shown. Additionally, we also plot a pure Gaussian kernel with variance $\omega^2$, scaled to match the maximum and minimum values of the NTK. We can observe the NTK kernel closely matches the Gaussian. Moreover, we can also observe that, at $\tilde{x} = 0$ the maximum value is predicted by $k \approx \omega^2/2$, as expected from the scaling factors in the kernel in Equation 5.

This NTK suggests that training a simple sinusoidal network is approximately equivalent to performing kernel regression with a Gaussian kernel, a low-pass filter, with its bandwidth defined by $\omega$.

We note that even though this sinusoidal network kernel approximates a Gaussian kernel, an actual Gaussian kernel can be recovered if a combination of sine and cosine activations are employed, as demonstrated in Tsuchida (2020) (Proposition 18).

### D.3 DEEP SINUSOIDAL NETWORKS

We will now look at the full NNGP and NTK for sinusoidal networks of arbitrary depth. As we will see, due to the recursive nature of these kernels, for networks deeper than the ones analyzed in the previous section, their full unrolled expressions quickly become intractable intuitively, especially for the NTK. Nevertheless, these kernels can still provide some insight, into the behavior of their corresponding networks. Moreover, despite their symbolic complexity, we will also demonstrate empirically that the resulting kernels can be approximated by simple Gaussian kernels, even for deep networks.

#### D.3.1 SIMPLE SINUSOIDAL NETWORK

As demonstrated in the previous section, simple sinusoidal networks produce simpler NNGP and NTK kernels due to their Gaussian initialization. We thus begin this section by now analyzing SSNs first, starting with their general NNGP.

**Theorem 15.** *SSN NNGP. For a simple sinusoidal network with $L$ hidden layers $f^{(L)} : \mathbb{R}^{n_0} \to \mathbb{R}^{n_{L+1}}$ following Definition 1, as the size of the hidden layers $n_1, \ldots, n_L \to \infty$ sequentially, $f^{(L)}$*

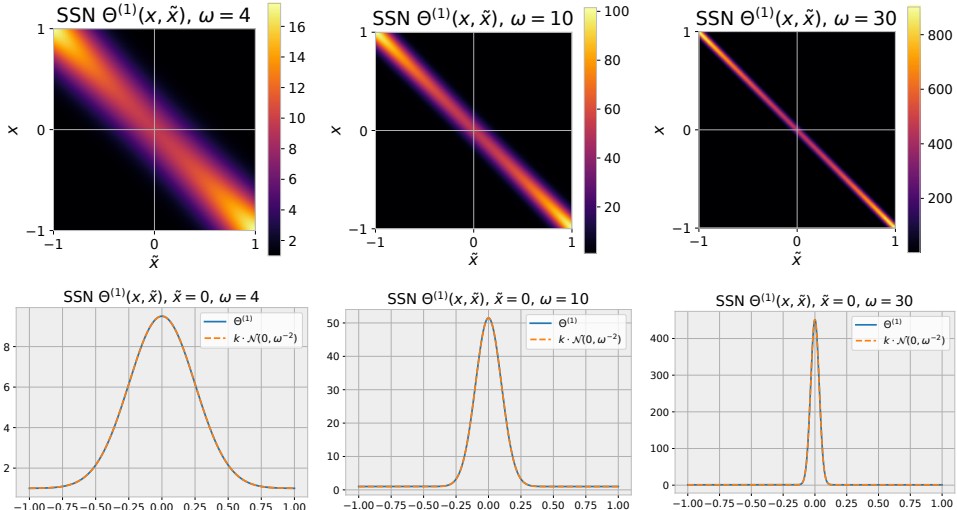

Figure 20: The NTK for SSN at different $\omega$ values. The top row shows the kernel values for pairs $(x, \tilde{x}) \in [-1, 1]^2$. Bottom row shows a slice at fixed $\tilde{x} = 0$, together with a Gaussian kernel scaled to match the maximum and minimum values of the NTK.

*tends (by law of large numbers) to the neural network Gaussian Process (NNGP) with covariance $\Sigma^{(L)}(x, \tilde{x})$, recursively defined as*

$$\Sigma^{(0)}(x, \tilde{x}) = \omega^2 \left( x^T \tilde{x} + 1 \right)$$

$$\Sigma^{(L)}(x, \tilde{x}) = \frac{1}{2} e^{-\frac{1}{2}\left(\Sigma^{(L-1)}(x,x) + \Sigma^{(L-1)}(\tilde{x},\tilde{x})\right)} \left( e^{\Sigma^{(L-1)}(x,\tilde{x})} - e^{-\Sigma^{(L-1)}(x,\tilde{x})} \right) + 1.$$

*Proof.* We will proceed by induction on the depth $L$, demonstrating the NNGP for successive layers as $n_1, \ldots, n_L \to \infty$ sequentially. To demonstrate the base case $L = 1$, let us rearrange $\Sigma^{(1)}$ from Theorem 12 in order to express it in terms of inner products,

$$\Sigma^{(1)}(x, \tilde{x}) = \frac{1}{2} \left( e^{-\frac{\omega^2}{2}\|x - \tilde{x}\|_2^2} + e^{-\frac{\omega^2}{2}\|x + \tilde{x}\|_2^2} e^{-2\omega^2} \right) + 1$$

$$= \frac{1}{2} \left[ e^{-\frac{\omega^2}{2}\left(x^T x - 2x^T \tilde{x} + \tilde{x}^T \tilde{x}\right)} - e^{-\frac{\omega^2}{2}\left(x^T x + 2x^T \tilde{x} + \tilde{x}^T \tilde{x}\right)} e^{-2\omega^2} \right] + 1$$

$$= \frac{1}{2} \left[ e^{-\frac{1}{2}\left[\omega^2\left(x^T x + 1\right) + \omega^2\left(\tilde{x}^T \tilde{x} + 1\right)\right] + \omega^2\left(x^T \tilde{x} + 1\right)} - e^{-\frac{1}{2}\left[\omega^2\left(x^T x + 1\right) + \omega^2\left(\tilde{x}^T \tilde{x} + 1\right)\right] - \omega^2\left(x^T \tilde{x} + 1\right)} \right] + 1.$$

Given the definition of $\Sigma^{(0)}$, this is equivalent to

$$\frac{1}{2} e^{-\frac{1}{2}\left(\Sigma^{(0)}(x,x) + \Sigma^{(0)}(\tilde{x},\tilde{x})\right)} \left( e^{\Sigma^{(0)}(x,\tilde{x})} - e^{-\Sigma^{(0)}(x,\tilde{x})} \right) + 1,$$

which concludes this case.

Now given the inductive hypothesis, as $n_1, \ldots, n_{L-1} \to \infty$ we have that the first $L-1$ layers define a network $f^{(L-1)}$ with NNGP given by $\Sigma^{(L-1)}(x, \tilde{x})$. Now it is left to show that as $n_L \to \infty$, we get the NNGP given by $\Sigma^{(L)}$. Following the same argument in Theorem 12, the network

$$f^{(L)}(x) = W^{(L)} \frac{1}{\sqrt{n_L}} \sin\left( f^{(L-1)} \right) + b^{(L)}$$

constitutes a Gaussian process given the outputs of the previous layer, due to the distributions of $W^{(L)}$ and $b^{(L)}$. Its covariance is given by $\sigma_W^2 \Sigma^{(L)} + \sigma_b^2 = \Sigma^{(L)} + 1$, where

$$\Sigma^{(L)}(x, \tilde{x}) = \lim_{n_L \to \infty} \left[ \frac{1}{n_L} \left\langle \sin\left( f^{(L-1)}(x) \right), \sin\left( f^{(L-1)}(\tilde{x}) \right) \right\rangle \right]$$

$$= \lim_{n_L \to \infty} \left[ \frac{1}{n_L} \sum_{j=1}^{n_L} \sin\left( f^{(L-1)}(x) \right)_j \sin\left( f^{(L-1)}(\tilde{x}) \right)_j \right].$$

By inductive hypothesis, $f^{(L-1)}$ is a Gaussian process $\Sigma^{(L-1)}(x, \tilde{x})$. Thus by the LLN the limit above equals

$$\mathbb{E}_{(u,v)\sim\mathcal{N}\left(0,\Sigma^{(L-1)}(x,\tilde{x})\right)} \left[\sin(u)\sin(v)\right].$$

Omitting the distribution from the expectation for brevity and expanding the exponential definition of sine, we have

$$\mathbb{E}\left[\frac{1}{2i}\left(e^{iu} - e^{-iu}\right)\frac{1}{2i}\left(e^{iv} - e^{-iv}\right)\right] = -\frac{1}{4}\left[\mathbb{E}\left[e^{i(u+v)}\right] - \mathbb{E}\left[e^{i(u-v)}\right] - \mathbb{E}\left[e^{-i(u-v)}\right] + \mathbb{E}\left[e^{-i(u+v)}\right]\right].$$

Since $u$ and $v$ are jointly Gaussian, $p = u + v$ and $m = u - v$ are also Gaussian, with mean 0 and variance

$$\sigma_p^2 = \sigma_u^2 + \sigma_v^2 + 2\operatorname{Cov}[u,v] = \Sigma^{(L-1)}(x,x) + \Sigma^{(L-1)}(\tilde{x},\tilde{x}) + 2\,\Sigma^{(L-1)}(x,\tilde{x}),$$
$$\sigma_m^2 = \sigma_u^2 + \sigma_v^2 - 2\operatorname{Cov}[u,v] = \Sigma^{(L-1)}(x,x) + \Sigma^{(L-1)}(\tilde{x},\tilde{x}) - 2\,\Sigma^{(L-1)}(x,\tilde{x}).$$

We can now rewriting the expectations in terms of normalized variables

$$-\frac{1}{4}\left[\mathbb{E}_{z\sim\mathcal{N}(0,1)}\left[e^{i\sigma_p z}\right] - \mathbb{E}_{z\sim\mathcal{N}(0,1)}\left[e^{i\sigma_m z}\right] - \mathbb{E}_{z\sim\mathcal{N}(0,1)}\left[e^{-i\sigma_m z}\right] + \mathbb{E}_{z\sim\mathcal{N}(0,1)}\left[e^{-i\sigma_p z}\right]\right].$$

Applying Proposition 7 to each expectation, we get

$$\frac{1}{2}\left[e^{-\frac{1}{2}\sigma_m^2} - e^{-\frac{1}{2}\sigma_p^2}\right]$$
$$= \frac{1}{2}\left[e^{-\frac{1}{2}\left(\Sigma^{(L-1)}(x,x)+\Sigma^{(L-1)}(\tilde{x},\tilde{x})-2\,\Sigma^{(L-1)}(x,\tilde{x})\right)} - e^{-\frac{1}{2}\left(\Sigma^{(L-1)}(x,x)+\Sigma^{(L-1)}(\tilde{x},\tilde{x})+2\,\Sigma^{(L-1)}(x,\tilde{x})\right)}\right]$$
$$= \frac{1}{2}e^{-\frac{1}{2}\left(\Sigma^{(L-1)}(x,x)+\Sigma^{(L-1)}(\tilde{x},\tilde{x})\right)}\left(e^{\Sigma^{(L-1)}(x,\tilde{x})} - e^{-\Sigma^{(L-1)}(x,\tilde{x})}\right)$$

$\square$

Unrolling the definition beyond $L = 1$ leads to expressions that are difficult to parse. However, without unrolling, we can rearrange the terms in the NNGP above as

$$\Sigma^{(L)}(x,\tilde{x}) = \frac{1}{2}e^{-\frac{1}{2}\left(\Sigma^{(L-1)}(x,x)+\Sigma^{(L-1)}(\tilde{x},\tilde{x})\right)}\left(e^{\Sigma^{(L-1)}(x,\tilde{x})} - e^{-\Sigma^{(L-1)}(x,\tilde{x})}\right) + 1$$
$$= \frac{1}{2}\left[e^{-\frac{1}{2}\left(\Sigma^{(L-1)}(x,x)-2\Sigma^{(L-1)}(x,\tilde{x})+\Sigma^{(L-1)}(\tilde{x},\tilde{x})\right)} - e^{-\frac{1}{2}\left(\Sigma^{(L-1)}(x,x)+2\Sigma^{(L-1)}(x,\tilde{x})+\Sigma^{(L-1)}(\tilde{x},\tilde{x})\right)}\right] + 1.$$

Since the covariance *matrix* $\Sigma^{(L-1)}$ is positive semi-definite, we can observe that the exponent expressions can be reformulated into a quadratic forms analogous to the ones in Theorem 12. We can thus observe that the same structure is essentially preserved through the composition of layers, except for the $\omega$ factor present in the first layer. Moreover, given this recursive definition, since the NNGP at any given depth $L$ is a function only of the preceding kernels, the resulting kernel will also be shift-invariant.

Let us now derive the $\dot{\Sigma}$ kernel, required for the NTK.

**Lemma 16.** *For $\omega \in \mathbb{R}$, $\dot{\Sigma}^{(L)}(x,\tilde{x}) : \mathbb{R}^{n_0} \times \mathbb{R}^{n_0} \to \mathbb{R}$, is given by*

$$\dot{\Sigma}^{(L)}(x,\tilde{x}) = \frac{1}{2}e^{-\frac{1}{2}\left(\Sigma^{(L-1)}(x,x)+\Sigma^{(L-1)}(\tilde{x},\tilde{x})\right)}\left(e^{\Sigma^{(L-1)}(x,\tilde{x})} + e^{-\Sigma^{(L-1)}(x,\tilde{x})}\right) + 1.$$

*Proof.* The proof follows the same pattern as Theorem 15, with the only difference being a few sign changes after the exponential expansion of the trigonometric functions, due to the different identities for sine and cosine. $\square$

As done in the previous section, it would be simple to now derive the full NTK for a simple sinusoidal network of arbitrary depth by applying Theorem 6 with the NNGP kernels from above. However,

there is not much to be gained by writing the convoluted NTK expression explicitly, beyond what we have already gleaned from the NNGP above.

Nevertheless, some insight can be gained from the recursive expression of the NTK itself, as defined in Theorem 6. First, note that, as before, for practical values of $\omega$, $\dot{\Sigma} \approx \Sigma$, both converging to simply a single Gaussian kernel. Thus, our NTK recursion becomes

$$\Theta^{(L)}(x, \tilde{x}) \approx \left(\Theta^{(L-1)}(x, \tilde{x}) + 1\right)\Sigma^{(L)}(x, \tilde{x}).$$

Now, note that when expanded, the form of this NTK recursion is essentially as a product of the Gaussian $\Sigma$ kernels,

$$\Theta^{(L)}(x, \tilde{x}) \approx \left(\left(\ldots\left(\left(\Sigma^{(0)}(x, \tilde{x}) + 1\right)\Sigma^{(1)}(x, \tilde{x}) + 1\right)\ldots\right)\Sigma^{(L-1)}(x, \tilde{x}) + 1\right)\Sigma^{(L)}(x, \tilde{x})$$
$$= \left(\left(\ldots\left(\left(\omega^2\left(x^T\tilde{x} + 1\right) + 1\right)\Sigma^{(1)}(x, \tilde{x}) + 1\right)\ldots\right)\Sigma^{(L-1)}(x, \tilde{x}) + 1\right)\Sigma^{(L)}(x, \tilde{x}).$$
(6)

We know that the product of two Gaussian kernels is Gaussian and thus the general form of the kernel should be approximately a sum of Gaussian kernels. As long as the magnitude of one of the terms dominates the sum, the overall resulting kernel will be approximately Gaussian. Empirically, we observe this to be the case, with the inner term containing $\omega^2$ dominating the sum, for reasonable values (*e.g.*, $\omega > 1$ and $L < 10$). In Figure 21, we show the NTK for networks of varying depth and $\omega$, together with a pure Gaussian kernel of variance $\omega^2$, scaled to match the maximum and minimum values of the NTK. We can observe that the NTKs are still approximately Gaussian, with their maximum value approximated by $k \approx \frac{1}{2^L}\omega^2$, as expected from the product of $\omega^2$ and $L$ kernels above. We also observe that the width of the kernels is mainly defined by $\omega$.

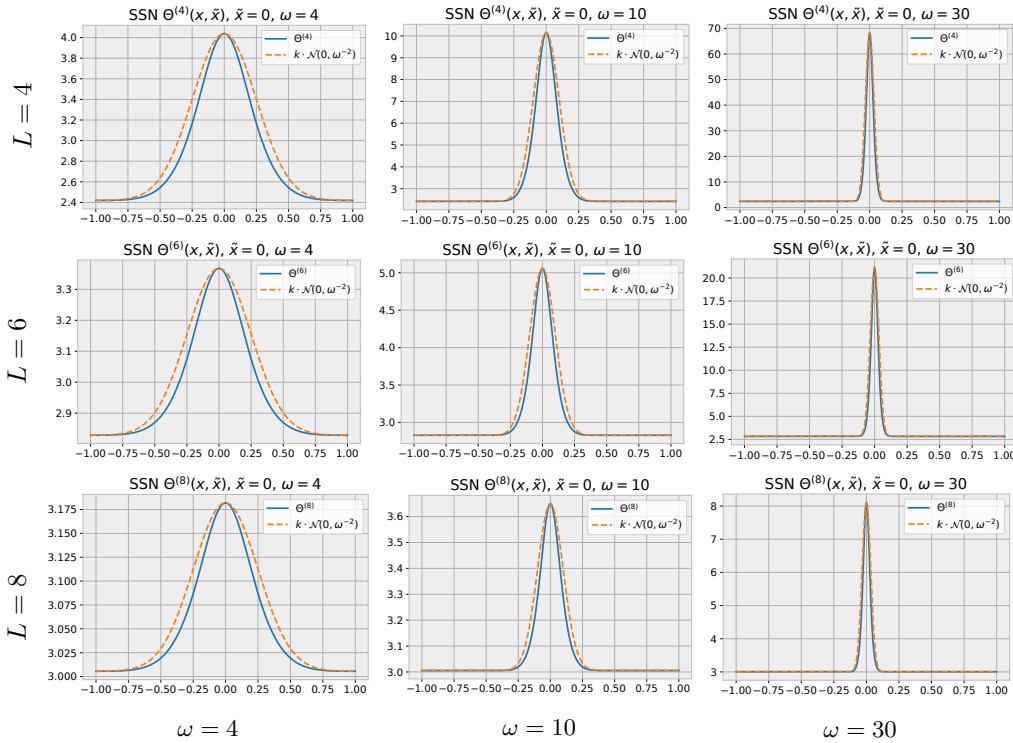

Figure 21: The NTK for SSN at different $\omega$ and network depth ($L$) values. Kernel values at a slice for fixed $\tilde{x} = 0$ are shown, together with a Gaussian kernel scaled to match the maximum and minimum values of the NTK.

### D.3.2   SIREN

For completeness, in this section we will derive the full SIREN NNGP and NTK. As discussed previously, both the SIREN and the simple sinusoidal network have kernels that approximate low-

pass filters. Due to the SIREN initialization, its NNGP and NTK were previously shown to have more complex expressions. However, we will show in this section that the sinc kernel that arises from the shallow SIREN is gradually "dampened" as the depth of the network increases, gradually approximating a Gaussian kernel.

**Theorem 17. *SIREN NNGP. For a SIREN with $L$ hidden layers $f^{(L)} : \mathbb{R}^{n_0} \to \mathbb{R}^{n_{L+1}}$ following Definition 1, as the size of the hidden layers $n_1, \ldots, n_L \to \infty$ sequentially, $f^{(L)}$ tends (by law of large numbers) to the neural network Gaussian Process (NNGP) with covariance $\Sigma^{(L)}(x, \tilde{x})$, recursively defined as***

$$\Sigma^{(1)}(x, \tilde{x}) = \frac{c^2}{6} \left[ \prod_{j=1}^{n_0} \mathrm{sinc}(c\,\omega\,(x_j - \tilde{x}_j)) - e^{-2\omega^2} \prod_{j=1}^{n_0} \mathrm{sinc}(c\,\omega\,(x_j + \tilde{x}_j)) \right] + 1$$

$$\Sigma^{(L)}(x, \tilde{x}) = \frac{1}{2} e^{-\frac{1}{2}\left(\Sigma^{(L-1)}(x,x) + \Sigma^{(L-1)}(\tilde{x},\tilde{x})\right)} \left( e^{\Sigma^{(L-1)}(x,\tilde{x})} - e^{-\Sigma^{(L-1)}(x,\tilde{x})} \right) + 1.$$

*Proof.* Intuitively, after the first hidden layer, the inputs to every subsequent hidden layer are of infinite width, due to the NNGP assumptions. Therefore, due to the CLT, the pre-activation values at every layer are Gaussian, and the NNGP is unaffected by the uniform weight initialization (compared to the Gaussian weight initialization case). The only layer for which this is not the case is the first layer, since the input size is fixed and finite. This gives rise to the different $\Sigma^{(1)}$.

Formally, this proof proceed by induction on the depth $L$, demonstrating the NNGP for successive layers as $n_1, \ldots, n_L \to \infty$ sequentially. The base case comes straight from Theorem 9. After the base case, the proof follows exactly the same as in Theorem 15. □

For the same reasons as in the proof above, the $\dot{\Sigma}$ kernels after the first layer are also equal to the ones for the simple sinusoidal network, given in Lemma 16.

Given the similarity of the kernels beyond the first layer, the interpretation of this NNGP is the same as discussed in the previous section for the simple sinusoidal network.

Analogously to the SSN case before, the SIREN NTK expansion can also be approximated as a product of $\Sigma$ kernels, as in Equation 6. The product of a sinc function with $L-1$ subsequent Gaussians "dampens" the sinc, such that as the network depth increases the NTK approaches a Gaussian, as can be seen in Figure 22.

# E    EXPERIMENTAL DETAILS

## E.1    GENERALIZATION

Where not explicitly commented, details for the generalization experiments are the same for the comparisons to SIREN, described in Appendix C.

## E.2    BURGERS EQUATION (IDENTIFICATION)

We follow the same training procedures as in Raissi et al. (2019a). The training set is created by randomly sampling $2{,}000$ points from the available exact solution grid (shown in Figure 5). The neural networks used are 9-layer MLPs with 20 neurons per hidden layer. The network structure is the same for both the tanh and sinusoidal networks. As in the original work, the network is trained by using L-BFGS to minimize a mean square error loss composed of the sum of an MSE loss over the data points and a physics-informed MSE loss derived from the equation

$$u_t + \lambda_1 u u_x - \lambda_2 u_{xx} = 0. \tag{7}$$

## E.3    NAVIER-STOKES (IDENTIFICATION)

We follow the same training procedures as in Raissi et al. (2019a). The training set is created by randomly sampling $5{,}000$ points from the available exact solution grid (one timestep is shown in

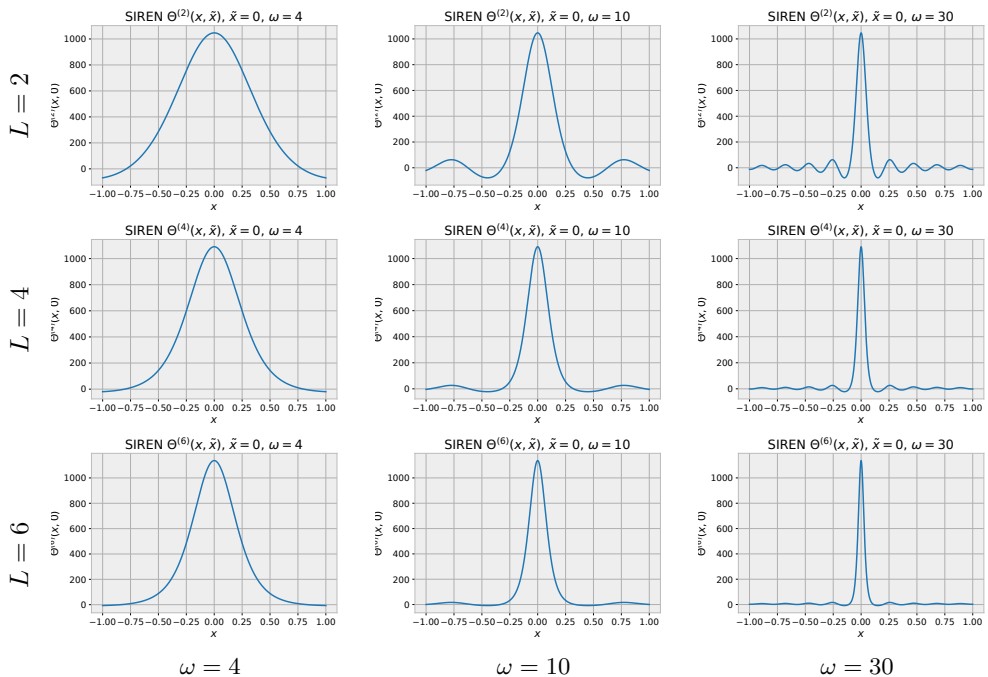

Figure 22: The NTK for SIREN at different $\omega$ and network depth ($L$) values. Kernel values at a slice for fixed $\tilde{x} = 0$ are shown.

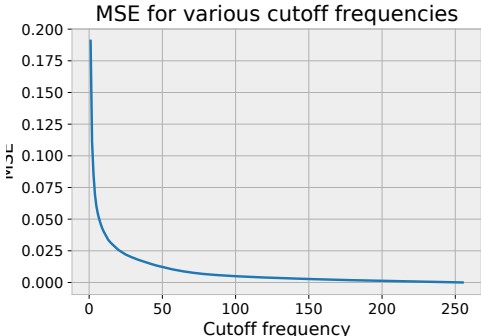

Figure 23: Reconstruction loss through a low-pass filter for the "camera" image. Notice the loss continues to go down up to the Nyquist frequency.

Figure 6). The neural networks used are 9-layer MLPs with 20 neurons per hidden layer. The network structure is the same for both the $\tanh$ and sinusoidal networks. As in the original work, the network is trained by using the Adam optimizer to minimize a mean square error loss composed of the sum of an MSE loss over the data points and a physics-informed MSE loss derived from the equations

$$u_t + \lambda_1(uu_x + vu_y) = -p_x + \lambda_2(u_{xx} + u_{yy}) \tag{8}$$
$$v_t + \lambda_1(uv_x + vv_y) = -p_y + \lambda_2(v_{xx} + v_{yy}). \tag{9}$$

### E.4 SCHRÖDINGER (INFERENCE)

This experiment reproduces the Schrödinger equation experiment from Raissi et al. (2019a). In this experiment, we are trying to find the solution to the Schrödinger equation, given by

$$ih_t + 0.5h_{xx} + |h|^2 h = 0 \tag{10}$$

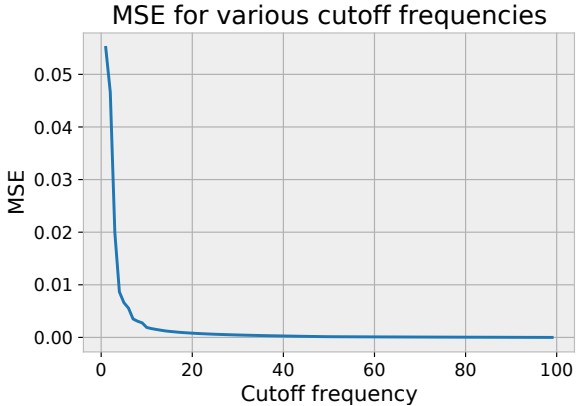

Figure 24: Reconstruction loss for different cutoff frequencies for a low-pass filter applied to the solution of the Navier-Stokes equations.

Since in this case we have a forward problem, we do not have any prior information to base our choice of $\omega$ on, besides a maximum limit given by the Nyquist frequency given the sampling for our training data. We thus follow usual machine learning procedures and experiment with a number of small $\omega$ values, based on the previous experiments.

We find that $\omega = 4$ gives the best results, with a solution MSE of $4.30 \cdot 10^{-4}$, against an MSE of $1.04 \cdot 10^{-3}$ for the baseline. Figure 25 shows the solution from the sinusoidal network, together with the position of the sampled data points used for training.

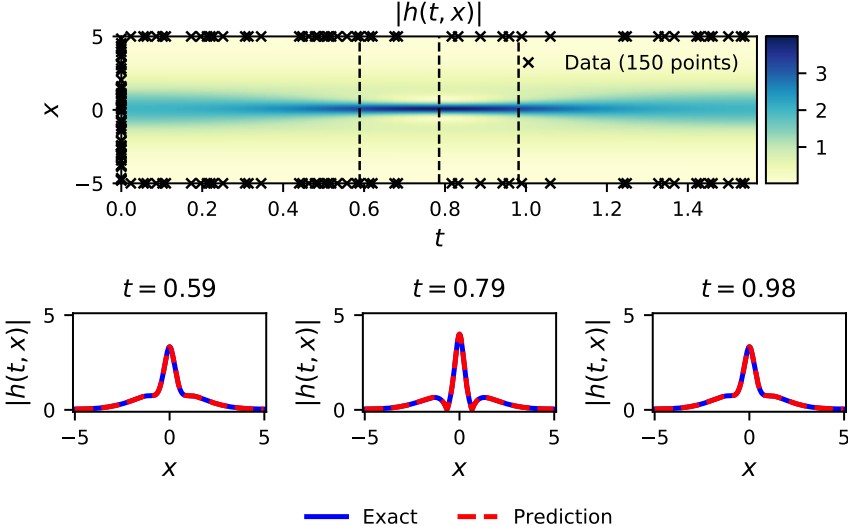

Figure 25: Solution to the Schrodinger equation with the sinusoidal network, together with the position of the sampled data points used for training.

**Training details.** We follow the same training procedures as in Raissi et al. (2019a). The training set is created by randomly sampling $20,000$ points from the domain ($x \in [-5, 5]$, $t \in [0, \pi/2]$) for evaluation for the physics-informed loss. Additionally, $50$ points are sampled from each of the boundary and initial conditions for direct data supervision. The neural networks used are 5-layer MLPs with 100 neurons per hidden layer. The network structure is the same for both the $\tanh$ and sinusoidal networks. As in the original work, the network is trained first using the Adam optimizer by $50,000$ steps and then by using L-BFGS until convergence. The loss is composed of the sum of an MSE loss over the data points and a physics-informed MSE loss derived from Equation 10.

### E.5 HELMHOLTZ EQUATION

Details for the Helmholtz equation experiment are the same as in the previous Helmholtz experiment in Appendix C.

