# OpenReview forum: "Simple initialization and parametrization of sinusoidal networks via their kernel bandwidth"
_ICLR.cc/2023/Conference — ICLR 2023 poster_

### Official Review · Reviewer_YnQa · 2022-10-24

**Confidence:** 2
**Correctness:** 4
**Technical Novelty And Significance:** 4
**Empirical Novelty And Significance:** Not applicable
**Recommendation:** 8

**Clarity, Quality, Novelty And Reproducibility:**

This is a well-written paper.  Sinusoidal neural networks are increasingly being used to learn implicit models, and the work presented in this paper sheds light on the inner working on these networks.  In addition, the work also presents guidelines that can help an interested reader design sinusoidal networks that exhibit better performance and achieve faster convergence.  This is all good news.

**Strength And Weaknesses:**

This is a well-written paper.  I quite liked the narrative structure of this manuscript.

The paper begins by constructing a simplified sinusoidal network model that mimics the key characterstics of SIREN but is much more amenable to theoretical analysis.  In addition to providing mathematical reasoning that confirms that the simplified sinusoidal model used in this work is similar to SIREN, the paper also provides empirical results that show that the simplified network achieves performance similar to that attained by SIREN.

Section 4 shows that the kernals of the simplified network approximates a Gaussian kernel whose width can be tuned.

Section 5 uses a toy example to show how network behaves as this "width" parameter value shifts.  It is not immediately obvious to me how to parse the results presented in Figure 3.  The following sentence confuses me, "We can see that due to the simple nature of the signal, containing only two frequencies, there are only three loss levels."  Why is this?  It would be useful for a reader like me to include a sentence for why there are only three loss levels for the current two-frequency problem setup.

Section 6 discusses how to tune the aforementioned "width" parameter.  The motivation being that this value is "crucial for the learning of the network."  This discussion presents a heuristic for setting this "width" parameter to one-eighth of the maximum frequency in the signal.  Results in Figures 3 seems to imply this heuristic.  This section of the paper is somewhat underwhelming.  Why one-eighth?  Why not, say, one-tenth?  Given that it is okay to select a slightly suboptimal value for this "width" parameter since the network is able to adjust it during training.  It is however clear that using too large a value may results in overfitting.

Section 7 and 8 presents results and conclusions.

**Summary Of The Paper:**

The paper analyzes sinusoidal neural networks from the Neural Tanget Kernal (NTK) perspective.  This analysis leads to a number of important observations.  The most important finding, perhaps, is that their NTK approximates a tuneable low-pass filter.  This insight is subsequently used to develop guidelines for optimizing the performance of a sinusoidal network by tuning the bandwidths of its kernels according the maximum frequency present in the input signal.  The paper also suggests an initialization scheme for sinusoidal networks that leads to improved results.  The ideas developed in this work are evaluated using two tasks---1) learning implicit models and 2) solving differential equations---and the results suggest that the ideas developed in this work have merit.

**Summary Of The Review:**

This paper advances our understanding of sinusoidal networks, consequently this paper will be of interest to the larger machine learning community, and in particular the ICLR community.

---

> ### Author Response · Authors · 2022-11-14
> **Response**
>
> We thank the reviewer for the careful analysis, for the detailed comments, and for the positive words about our work. Below we respond to the questions presented.
>
>
> *“The following sentence confuses me…”*
>
> That sentence refers to performing a reconstruction of the image after filtering it with an ideal low-pass filter. Since the original image is constructed from only 2 frequencies, there are only 3 possible cases: (1) the cutoff frequency for the low-pass filter is lower than the 2 frequencies, in which no frequency passes through, and the reconstruction achieves a high loss level, (2) the cutoff frequency for the filter is in between the two frequencies, in which case the lower frequency passes and the reconstruction achieves an intermediate loss, and (3) the cutoff frequency is above the highest of the 2 original frequencies, and thus both frequencies are used to reconstruct the image, which can be fully reconstructed and thus a loss of 0 is achieve. As there are only these 3 cases, we should have 3 loss levels, when using an ideal low pass filter (which is not the case for the neural network, but this sentence was referring only to the Fourier transform).
>
>
> *“Why one-eighth? Why not, say, one-tenth?...”*
>
> Indeed, as mentioned in the paper, this value is a heuristic guide and is not intended to be interpreted as a hard-set rule. We use the “one-eight” value as the proposed value because that is the value that showed up in our experiments as best performing and also, of course, because it is a nice round number. Given their proximity and the fact that there is some latitude when setting this parameter, it is probable that one-tenth, or one-ninth, would also work approximately as well.

---

### Official Review · Reviewer_FbCm · 2022-10-24

**Confidence:** 3
**Correctness:** 3
**Technical Novelty And Significance:** 2
**Empirical Novelty And Significance:** 3
**Recommendation:** 6

**Clarity, Quality, Novelty And Reproducibility:**

The paper is well-written and easy to follow. The experiments are described in adequate detail from a reproducibility perspective. While the individual elements presented in this paper are not entirely novel, they are synthesised into a coherent and reasonably compelling story.

**Strength And Weaknesses:**

**Strengths:**
- To the best of my knowledge, this is the first paper that studies in-depth the way in which infinite-width neural network theory can help inform neural networks with sinusoidal activations that are trained using gradient-based optimisers.

**Mathematical correctness:**
- Theorem 2. In particular, the phrase "approximately standard normal distributed" is not precisely mathematically defined anywhere in the paper before the theorem is presented. Either change remove the theorem status of this statement, or define "approximately standard normal distributed". More importantly: asymptotic normality will hold (due to the previously cited results of Lee et al. 2018). If all you need is Lee et al.'s result, then just state this.
- "We can thus observe that this kernel approximates a Gaussian kernel, which is a low-pass filter, with its bandwidth define by ω". If you really want a squared exponential, RBF kernel, you can obtain this using half sine and half cosine activations. You can observe that summing the two kernels cancels out some terms. For example, see Proposition 18 of "Results on Infinitely Wide Multi-layer Perceptrons".

**Minor:**
- Theorem 13 has typo in "shalow". Note this kernel is very similar to the kernel in "Expressive Priors in Bayesian Neural Networks: Kernel Combinations and Periodic Functions", which uses cosine instead of sine (but the analysis is pretty much identical). Maybe it is identical to Lemma 17. It is perhaps also worth mentioning that it is straight-forward to handle non-zero mean weights for these activations in the NNGP setting, since the integrals also have a closed-form expression. In general, this work might be worth citing as a Bayesian counterpart to the gradient-descent approach considered here (with the other difference that cosine is replaced with sine, which is not too important).

**Summary Of The Paper:**

The authors analyse the NNGP kernel and NTK of neural networks with sinusoidal activations. The NTK has an adjustable band-width parameter, which allows users to understand such kernel models in terms of a low-pass filter. A discussion on how the cutoff frequency is chosen is provided. The authors show that such kernels can be applied to implicit models and differential equations.

**Summary Of The Review:**

This paper studies neural nets with sinusoidal activations in an NTK framework. A key bandwidth parameter and its importance on some representative problems is given. While the mathematical analysis does not require a huge amount of mathematical creativity, the paper is well organised and presents a clear story. There are some minor issues surrounding mathematical clarity and discussion of previous work.

---

> ### Author Response · Authors · 2022-11-14
> **Response**
>
> We thank the reviewer for the careful analysis, for the detailed comments, and for the positive words about our work. We have corrected the typo that was pointed out in the updated version of the manuscript.
>
> We also thank the reviewer for the suggestion of the related work, which, as mentioned, is an interesting Bayesian counterpart to our approach, with a greater focus on the NNGP setting. We will include this suggestion in the updated version of the paper here.
>
> Below we respond to the questions presented.
>
>
> *“Theorem 2…”*
>
> We agree with the problems pointed out in Theorem 2.
>
> Our intention with Theorem 2 was simply to show that the results that held for SIREN also hold for the SSN. As such, our Theorem 2 is essentially a reproduction of Theorem 1.8 from Sitzmann et al., while pointing out that all that is needed for the same argument to hold is for the variance to match what was specified in their argument (which is provided in Lemma 1).
>
> However, the proof from Sitzmann et al. for their Theorem 1.8 indeed has the of the approximation to the standard normal distribution which are not formalized, as you point out. We will thus accept the suggestion and address this problem with Sitzmann et al.’s proof by: (1) pointing out the issues with the original Theorem, noting that it is not in fact a formal statement, and (2) removing the claim that our result is a Theorem, and simply state that the informal argument (with empirical support) from Sitzmann et al. still holds for the case of SSNs as long as the variance is matched between the networks. We believe this maintains the original point we wanted to make (that SSNs can be made essentially equivalent to SIRENs by choosing their initialization appropriately), while avoiding the problems with the Sitzmann et al. Theorem 1.8 that were carefully observed by the reviewer.
>
> Finally, we also note that this result is in no way central to any of the main results in our paper, it being only a side observation that SSNs can be made equivalent to SIRENs with the appropriate variance. Even if this result did not hold for some reason, all the other analyses of SSNs and SIRENs we perform would individually still hold.
>
>
> *“If you really want a squared exponential, RBF kernel, you can obtain this using half sine and half cosine activations…”*
>
> We thank the reviewer for the suggestion. Indeed we had considered that framing, yet ultimately refrained from using it due to the fact that it would move us away from the most practical applications of sinusoidal networks, such as SIRENs, which simply use sine activations. As with any work of this kind, we tried to balance out the most interesting theoretical approach with keeping the framing close to what is commonly used in practice. We ended up deciding to compromise here by using purely sinusoidal networks. Nevertheless, we agree that this is indeed an interesting point to make, and will add commentary noting this possibility in the updated version. (It is possible that such addition will have to be made in the Appendix due to space constraints).

---

> > ### Comment · Reviewer_FbCm · 2022-11-17
> > **Thanks for your response**
> >
> > Thanks for responding to my comments. Your response was satisfactory, and I look forward to seeing the updated version. In future, I would encourage the authors to take advantage of the fact that ICLR allows revised manuscripts to be posted. This would be very helpful in particular around Theorem 2, for which sbxp also had separate feedback.

---

> > > ### Author Response · Authors · 2022-11-19
> > > **Response**
> > >
> > > We have added the updated version addressing the comments from all reviewers. Thanks for the response.

---

### Official Review · Reviewer_pDxg · 2022-10-27

**Confidence:** 3
**Correctness:** 4
**Technical Novelty And Significance:** 3
**Empirical Novelty And Significance:** 3
**Recommendation:** 8

**Clarity, Quality, Novelty And Reproducibility:**

Quality and Clarity: Great. The writing and organization of this paper is clear and well structured. The basic idea and methodology is easy to understand by readers, and the theoretical analysis and its related framework were solid.

Originality: Good. This paper is mainly based on two existing foundations, sinusoidal network and NTK technology. The author modified the original sinusoidal network to make it easy to analyze and outperform the original one empirically. Most computation of the NTK analysis used the existing framework of NTK, but the results are interesting and striking enough. Besides, the theoretical results match exactly with the experiments, endorsing the correctness of theoretical analysis.

**Strength And Weaknesses:**

Strength:
- The modification over the original SIREN networks is simple, effective and easier to analyze.
- The theoretical framework is solid and clear.
- It is a significant discovery that the sinusoidal networks work approximately as low-pass filters with bandwidth controlled by w, which can be utilized to design the initialization parameters of sinusoidal networks.

Weakness:

- This paper compared the Simple Sinusoidal Network with the original implementation of SIREN using different initialization strategies. I wonder how much contribution the Kaiming normal initialization and different network models make respectively.
- What happens if applying the proposed initialization method to the original sinusoidal networks? Will it behave whether a comparable or a worse performance?
- One question: You claimed that the modified sinusoidal networks are easier to analyze in theoretical way, but in Corollary 12 it seems that it is possible to figure out the NTK expression for original SIREN networks as well. In what aspect the modified model could embody better potential for analysis?

**Summary Of The Paper:**

1. This paper proposed a simplified version of the original form of sinusoidal networks in SIREN, that allows for easier implementation and theoretical analysis.
2. The proposed sinusoidal networks outperform SIRENs in implicit representation learning tasks.
3. Analysis of the proposed sinusoidal networks from an NTK perspective. The NTK approximates a low-pass filter with adjustable bandwidth.
4. Demonstrate that the performance of a sinusoidal network can be optimized by tuning the bandwidth implied in the NTK approximation.

**Summary Of The Review:**

The message that this paper delivers is simple but helpful to understanding the behavior of sinusoidal networks. It also provides some guidelines of choosing better initialization parameters to achieve better models empirically. In the meantime, from the technical perspective, it is a successful attempt to use NTK as an approximation technology to analyze the proposed model in the theoretical way.

---

> ### Author Response · Authors · 2022-11-14
> **Response**
>
> We thank the reviewer for the careful analysis, for the detailed comments, and for the positive words about our work. Below we respond to the question presented.
>
>
> *“I wonder how much contribution the Kaiming normal initialization and different network models make respectively.”*
>
> In our experiments, switching either the SSN or SIREN between normal or uniform Kaiming initializations did not impact performance significantly either way, as along as initialization variance was preserved, as we perform in the paper.
>
>
> *“What happens if applying the proposed initialization method to the original sinusoidal networks? Will it behave whether a comparable or a worse performance?”*
>
> We will interpret here that by “original sinusoidal networks” you mean SIRENs. If you meant something different, feel free to reply and we can amend our response. Overall, we expect the same approach to work in SIRENs as well (up to possibly a difference in scalar multiplier). The main reasons why we propose and employ simple sinusoidal networks was because (1) of their simpler implementation while maintaining similar performance (which seems desirable) and (2) their somewhat theoretical analysis in this setting, as their NNGP and NTK end up with Gaussian forms. Nevertheless, we do provide analysis showing that the NTK for SIRENs is in the form of a sinc kernel, and should also work as a low-pass filter with bandwidth controlled by omega, and thus many of the same findings should apply.
>
>
> *“In what aspect the modified model could embody better potential for analysis?”*
>
> Even though we can derive the NNGPs and NTKs for SIRENs of arbitrary depth in closed form, we observed that while the kernels for these networks take the form of a sinc kernel for the single-layer (i.e., shallow) case, their form is more complex for deeper networks. This is described in more detail in Section D.3.2 of the Appendix, but essentially this  happens because while the first layer of the SIREN gives rise to the sinc kernel due to its uniform initialization, due to the CLT, all subsequent layers (which are of infinite width) follow essentially the same NTK derivation as in the normal initialization. Thus the kernel for the deep case ends up as a composition of the sinc kernel of the first layer, and the Gaussian kernel of the following layers. As you can see, even though it is possible to analyze this case, it can become more involved than the simpler, normal initialization one.

---

### Official Review · Reviewer_sbxp · 2022-10-29

**Confidence:** 2
**Correctness:** 3
**Technical Novelty And Significance:** 3
**Empirical Novelty And Significance:** 3
**Recommendation:** 6

**Clarity, Quality, Novelty And Reproducibility:**

## Clarity

The paper is very well written.

## Quality

As mentioned above, I have some doubts about Theorem 2 and the role of the scaling $\omega$.

Besides, I am not a specialist of periodic activation functions, so I cannot evaluate the impact of such theoretical results.

## Novelty

The results are definitely new.

**Strength And Weaknesses:**

## Strengths

The NTK computation for SIREN and SSN is new. It can definitely be useful to understand the dynamics of SIREN and SSN neural networks.

## Doubts

### Dependence on Sitzmann's proofs

Theorem 2 depends entirely on Sitzmann's proof (*Implicit Neural Representations with Periodic Activation Functions*, 2020). However, it appears that, by using their Theorem 1.5 (Central Limit Theorem), they assume implicitly that they are at the infinite-width limit, that is, the number of neurons per layer tends to infinity. So, this theorem is not valid with a finite number of neurons per layer. (Or I may have missed an argument.)

I recall that the tails of the distributions of the pre-activations in a ReLU NN tends to become heavier and heavier after each layer (in a finite-width setup). So, the term ``approximately standard normal'' should be more explicit: what is the considered distance?

### Scaling $\omega$

At the beginning of Section 3, the authors recall that, when initializing SIRENs, the weights are sampled from $\mathcal{U}([-c/\omega, c/\omega])$, excluding the weights of the first layer (why?). This choice is understandable in common NNs, and leads to different learning trajectories (while it does not change the function represented by the NN at initialization).

My questions are the following:
1. when initializing SSNs, the weights are Gaussian. But what is their variance?
2. it seems that, for SSNs and SIRENs, the weights are initialized without being scaled by $1/\omega$. Why?
3. if we scale the variance of the initialization distribution of the weights by $1/\omega$, do we recover similar results when computing the NTKs?

**Summary Of The Paper:**

This paper proposes a restriction of the SIREN structure (neural network with sine activation function), named SSN. SSN has a tunable frequency in the first layer only (instead of all the layers), and its weights are initialized with a Gaussian distribution, instead of a uniform distribution.

They compute and compare the Neural Tangent Kernel (NTK) of SIREN and SSN, and conclude that both perform low-pass filtering.

**Summary Of The Review:**

I have some doubts about two theoretical results, and I am not able to evaluate the impact of the results.

---

> ### Author Response · Authors · 2022-11-14
> **Response**
>
> We thank the reviewer for the careful analysis, for the detailed comments, and for the positive words about our work. Below we respond to the question presented.
>
>
> *“Theorem 2 depends entirely on Sitzmann's proof…”* and *“So, the term ``approximately standard normal'' should be more explicit: what is the considered distance?”*
>
> Indeed, Sitzmann’s proof has the issues you mention, both with the CLT assumption and the non-formal approximation of the standard normal distribution. We note that the approximations to the standard normal distribution are performed in Sitzmann et al. (see Lemma 1.6 in Sitzmann et all.), and are not formalized either.
>
> As you note, our claim depends entirely on Sitzmann et al.’s claim. Our intention with that Section of our paper was to simply show that the results that held for SIREN also hold for the SSN. However, we agree with the problems you point out, and will address it by: (1) pointing out the issues with the original Theorem, noting that it is not in fact a formal statement, and (2) removing the claim that our result is a Theorem, and simply state that the informal argument (with empirical support) from Sitzmann et al. still holds for the case of SSNs as long as the variance is matched between the networks. We believe this maintains the original point we wanted to make (that SSNs can be made essentially equivalent to SIRENs by choosing their initialization appropriately), while avoiding the problems with the Sitzmann et al. Theorem 1.8 that were carefully observed by the reviewer.
>
> We also note that this result is in no way central to any of the main results in our paper, it being only a side observation that SSNs can be made equivalent to SIRENs with the appropriate variance. Even if this result did not hold for some reason, all the other analyses of SSNs and SIRENs we perform would individually still hold.
>
>
> To your numbered questions:
> 1. *“when initializing SSNs, the weights are Gaussian. But what is their variance?”*
>
> The variance is 2/n, where n is the fan in.
>
> 2. *“it seems that, for SSNs and SIRENs, the weights are initialized without being scaled by 1/ω. Why?”*
>
> For the SSNs we propose, all layers after the first are not multiplied by omega, so there is no reason to scale them. For the first layer, which is the only one multiplied by omega, the point is precisely to have the omega act as a scaling factor, so that it can affect its spectrum, which is what our analyses demonstrate (and hinge on). If the first layer weights were initialized by being divided by omega, it would cancel out the multiplicative factor and likely negate the desired effect of omega.
> For SIRENs, the reasons seem to be similar. In practice, the implementation used in SIRENs has all layers multiplied by omega at inference time. In order to cancel that out af layers after the first one, the weight initialization is divided by omega (in the manner you propose). This essentially cancels the scaling by omega for the layers after the first one. For the first layers, this division by omega in the weight initialization is not applied, and it thus maintains the scaling from the forwards pass, having it also affect the spectrum, similarly to the described above for SSNs.
>
> 3. *“if we scale the variance of the initialization distribution of the weights by 1/ω, do we recover similar results when computing the NTKs?”*
>
> From the NTK perspective, scaling the weights during activation or scaling their variance is the same, and thus doing this proposed division by omega would cancel out the multiplication by omega that is performed intentionally (to change the spectrum of the kernel). Note that for SSNs only the first layer is multiplied by omega (see answer to question 2 above), so this canceling would happen only in the first layer (the other layers would just have their variance initialization arbitrarily divided by omega).

---

### Decision · Program_Chairs · 2023-01-20

**Decision:**

Accept: poster

**Justification For Why Not Higher Score:**

It would be interesting to investigate some more extensive applications of the initialization scheme to SIRENs, as proposed by reviewer pDxg.

**Justification For Why Not Lower Score:**

The narrative of the paper is very solid: motivation -> modeling -> analysis/insights -> feeding insights back to modeling (initialization) -> good empirical results. Well rounded paper that should be of interest to the community.

**Metareview: Summary, Strengths And Weaknesses:**

The authors derive a simplified sinusoidal network, the Simple Sinusoidal Network (SSN) in order to analyze such architectures from the NTK perspective, and use the generated insights to create an initialization strategy.

This work is in an interesting area that has enjoyed recent attention especially after the publication of Sitzmann et al. 2020. The work is, therefore, very timely.

Overall, the paper presents a well rounded and solid narrative: first, the creation of a simpler network for the purpose of analysis, with the additional theoretical demonstration that it can still be made equivalent to the original type of networks (SIREN). The analysis is novel and insightful. The insights feed back to the modeling step, to allow the authors to propose an effective initialization. The key strength is not only that the SSN ends up being very competitive despite its simplicity, but also that we now have a better understanding of sinusoidal networks as (in the NTK limit) are shown to approximate a low-pass filter with bandwidth that can be adjusted/optimized.

It would be interesting to investigate some more extensive applications of the initialization scheme to SIRENs, as proposed by reviewer pDxg. But in any case, the authors provide relevant theoretical analysis for SIRENs, so all in all the paper is certainly complete and will be useful to the community.

**Note From Pc:**

if the above contains the word "oral" or "spotlight" please see: "oral" presentation means -> notable-top-5% and "spotlight" means -> notable-top-25%. As stated in our emails, we are disassociating presentation type from AC recommendations